# *Arabidopsis* P4 ATPase-mediated cell detoxification confers resistance to *Fusarium graminearum* and *Verticillium dahliae*

Fanlong Wang [1], Xianbi Li[1], Yujie Li[1], Jing Han[1], Yang Chen[1], Jianyan Zeng[1], Mei Su[1], Jingxin Zhuo[1], Hui Ren[1], Haoru Liu[1], Lei Hou[1], Yanhua Fan[1], Xingying Yan[1], Shuiqing Song[1], Juan Zhao[1], Dan Jin[1], Mi Zhang [1] & Yan Pei [1]✉

Many toxic secondary metabolites produced by phytopathogens can subvert host immunity, and some of them are recognized as pathogenicity factors. Fusarium head blight and Verticillium wilt are destructive plant diseases worldwide. Using toxins produced by the causal fungi *Fusarium graminearum* and *Verticillium dahliae* as screening agents, here we show that the *Arabidopsis* P4 ATPases AtALA1 and AtALA7 are responsible for cellular detoxification of mycotoxins. Through AtALA1-/AtALA7-mediated vesicle transport, toxins are sequestered in vacuoles for degradation. Overexpression of *AtALA1* and *AtALA7* significantly increases the resistance of transgenic plants to *F. graminearum* and *V. dahliae*, respectively. Notably, the concentration of deoxynivalenol, a mycotoxin harmful to the health of humans and animals, was decreased in transgenic *Arabidopsis* siliques and maize seeds. This vesicle-mediated cell detoxification process provides a strategy to increase plant resistance against different toxin-associated diseases and to reduce the mycotoxin contamination in food and feed.

[1] The Affiliation Biotechnology Research Center, Southwest University, No. 2 Tiansheng Rd, Beibei, Chongqing 400716, P. R. China.
✉email: peiyan3@swu.edu.cn

Plant fungal diseases cause serious production and economic losses in agriculture and forestry worldwide. Many important phytopathogenic fungi, such as *Fusarium* species and *Verticillium* species, produce secondary metabolites that are toxic to host cells[1–3]. *F. graminearum* is the causal agent of Fusarium head blight (FHB) disease in cereals, which results in significant yield losses mainly in wheat, barley, and maize[4–11]. During pathogenic processes, the fungus produces epoxy-sesquiterpenoid compounds, named trichothecenes, including deoxynivalenol (DON), T-2 toxin, HT-2 toxin, and nivalenol[7,12]. These mycotoxins can evoke the rapid generation of reactive oxygen species that stimulate apoptosis-like processes and result in chlorotic and necrotic lesions, growth cessation, or even plant death[13–17]. Owing to their ability to inhibit protein synthesis, trichothecenes are also toxic to mammals. In fact, the contamination of trichothecene mycotoxins in food and feed commodities has become a global safety issue[18–20]. Verticillium wilt, mainly caused by the soil-borne vascular fungi *V. dahliae* and *V. albo-atrum*, is another destructive plant disease that results in tremendous economic losses worldwide. These diseases can be found in over 200 plant species, including many annual crops and perennial trees[21–31]. Owing to the complicated pathogenic mechanisms and the lack of host resistance to *F. graminearum*, successful control of FHB has been a major challenge for breeders[32,33]. Likewise, with a broad host range, high genetic diversity, and long-term persistence in the soil, *Verticillium* species have been ranked as some of the most difficult pathogens to control[27,30,34].

Mycotoxins have been regarded as the key virulence factors of FHB and Verticillium wilt[24,35]. Therefore, the detoxification of mycotoxins has been developed to control these formidable diseases. This strategy protects hosts from the toxicity of mycotoxins, thus allowing plants to maintain their innate immunity against the invasion of pathogens and increase their resistance to diseases caused by toxin producers.

In living cells, there are three major strategies for detoxification: (i) chemical modification/inactivation, (ii) compartmentation, and (iii) efflux[18]. At present, the main strategy used to detoxify trichothecenes in plant breeding is chemical modification/inactivation. UDP-glucosyltransferase (UGT) can conjugate DON into nontoxic DON-3-O-glucoside (D3G). DON resistance was achieved in *Arabidopsis*[36] and wheat[37] by overexpressing the *UGT* gene. In wheat, *Fhb1* was identified to contribute to resistance to a broad spectrum of *Fusarium* species[8,38–40]. Recently, it was found that *Fhb7* encodes a glutathione-S-transferase (GST) that could detoxify trichothecenes through de-epoxidation and thus confer broad resistance to *Fusarium* species[8]. However, the side effects on plant growth resulting from metabolic modification[36,41] and the toxicity reversal of the modified compounds[42,43] caused by this strategy remain to be solved.

*Verticillium* toxins include the high molecular weight protein lipopolysaccharide complex (PLPC)[44–46], small peptides, and low molecular weight metabolites[24,47,48]. Among them, low molecular weight metabolites are the key causal agents of wilt symptoms[24,48,49]. A lipophilic phytotoxin named cinnamate acetate (CIA) was identified from a strain of *V. dahliae* that is pathogenic to olive trees; this compound can induce Verticillium wilt-like symptoms and thus could be used to select *Verticillium*-tolerant olive trees[50]. Although *Verticillium* toxins have been used for the selection of disease-resistant plants[50,51], the targeted detoxification of these toxins has not been reported in plants due to the complicated composition of *Verticillium* toxins.

The P4 subfamily of P-type ATPases (P4 ATPases) functions as lipid flippases that translocate specific lipid substrates from the exoplasmic/luminal leaflet to the cytoplasmic leaflet of biological membranes, and these ATP-fueled flippases have a key role in the biogenesis of intracellular mobile vesicles[52–62]. Accumulating evidence in yeast, worms, plants and mammals has shown that P4 ATPase-mediated transport is involved in a variety of developmental and physiological processes, including the cell cycle, organelle biogenesis, embryogenesis, cellular polarization, hormone transport, and the response to biotic/abiotic stress[63–66]. In *Arabidopsis*, there are 12 members (AtALA1 to AtALA12; *Arabidopsis thaliana* aminophospholipid ATPases) in the P4 ATPase family. Although important roles of AtALAs have been reported in tolerance to temperature stresses[67–69], disease resistance[70–72], auxin polar distribution[73], cell expansion and plant growth[74,75], pollen and ovule development[76–78], little is known about the role of P4 ATPases in mycotoxin detoxification, even though ALA4 was demonstrated to be involved in heavy mental detoxification[79].

In this study, using the *Fusarium* toxin DON and *Verticillium* toxin CIA as probes, we identified two *Arabidopsis* P4 ATPase genes, *AtALA1* and *AtALA7* that were responsible for resistance to the toxins. We revealed that detoxification occurred via vesicle-mediated transport of toxins from the plasma membrane to vacuoles. Importantly, the expression of *AtALA7* in *Arabidopsis* and tobacco significantly increased the resistance to different strains of *V. dahliae*, whereas the exogenous overexpression of *AtALA1* in maize significantly increased the resistance to DON and ear rot disease, thus largely reducing the DON concentration in seeds. Our data demonstrated that this vesicle transport-mediated cell detoxification can be achieved without a growth penalty in different plant species targeting different toxins. Therefore, this strategy is promising for the development of resistance to recalcitrant diseases, e.g., FHB and Verticillium wilt, as well as for the reduction of mycotoxin contamination in food and feed.

## Results

**AtALA1 and AtALA7 are responsible for the vacuole compartmentation of DON and CIA.** After treatment of AtALA1 to 11 loss-of-function *Arabidopsis* mutants (Supplementary Fig. 1) with DON (50 μg/mL) and CIA (100 μg/mL), mutants for AtALA1 (*ala1*, *ala1-8*) and AtALA7 (*ala7*, *ala7-24*) showed significantly reduced tolerance to DON (Fig. 1a–c and Supplementary Fig. 2a, b) and CIA (Fig. 1f–h and Supplementary Fig. 2e, f), respectively. To investigate whether defects in AtALA1 and AtALA7 could affect the accumulation of toxins in vacuoles, we treated *ala1* and *ala7* mutants with DON labeled with 5-carboxyfluorescein (DON$^{5-FAM}$) and CIA labeled with fluorescein-isothiocyanate-isomer (CIA$^{FITC}$), which did not impair the toxicity of either DON (Supplementary Fig. 3a, b) or CIA (Supplementary Fig. 3c, d). With DON$^{5-FAM}$ treatment, no obvious fluorescent signal was observed in the vacuoles of cells of *ala1* mutant, whereas the signal appeared obvious in the vacuoles of other mutants (Fig. 1d, e and Supplementary Fig. 2c, d). Similarly, the CIA$^{FITC}$ signal was nearly undetectable in the *ala7* vacuoles, whereas the signal appeared obvious in the vacuoles of other mutants (Fig. 1i, j and Supplementary Fig. 2g, h). Moreover, the dynamic trafficking of DON$^{5-FAM}$ and CIA$^{FITC}$ in wild-type *Arabidopsis* and the mutants was observed. At 2 h, no DON$^{5-FAM}$-containing vesicles stained by FM4-64 were observed in root hair cells of the *ala1* mutant. However, the DON$^{5-FAM}$ signal was observed in the vesicles in root hair cells of wild-type and complemented plants (*proAtALA1::EGFP-AtALA1*, Comp1) (Fig. 1d). At 6 h, the DON$^{5-FAM}$ signal could be observed in small vacuoles of root meristem cells of wild-type and complemented plants; at 10 h, the signal accumulated in the mature vacuoles (Fig. 1d, e). In contrast, DON$^{5-FAM}$ appeared in the cytoplasm of the *ala1* mutant but not in vesicles or vacuoles (Fig. 1d). Similarly,

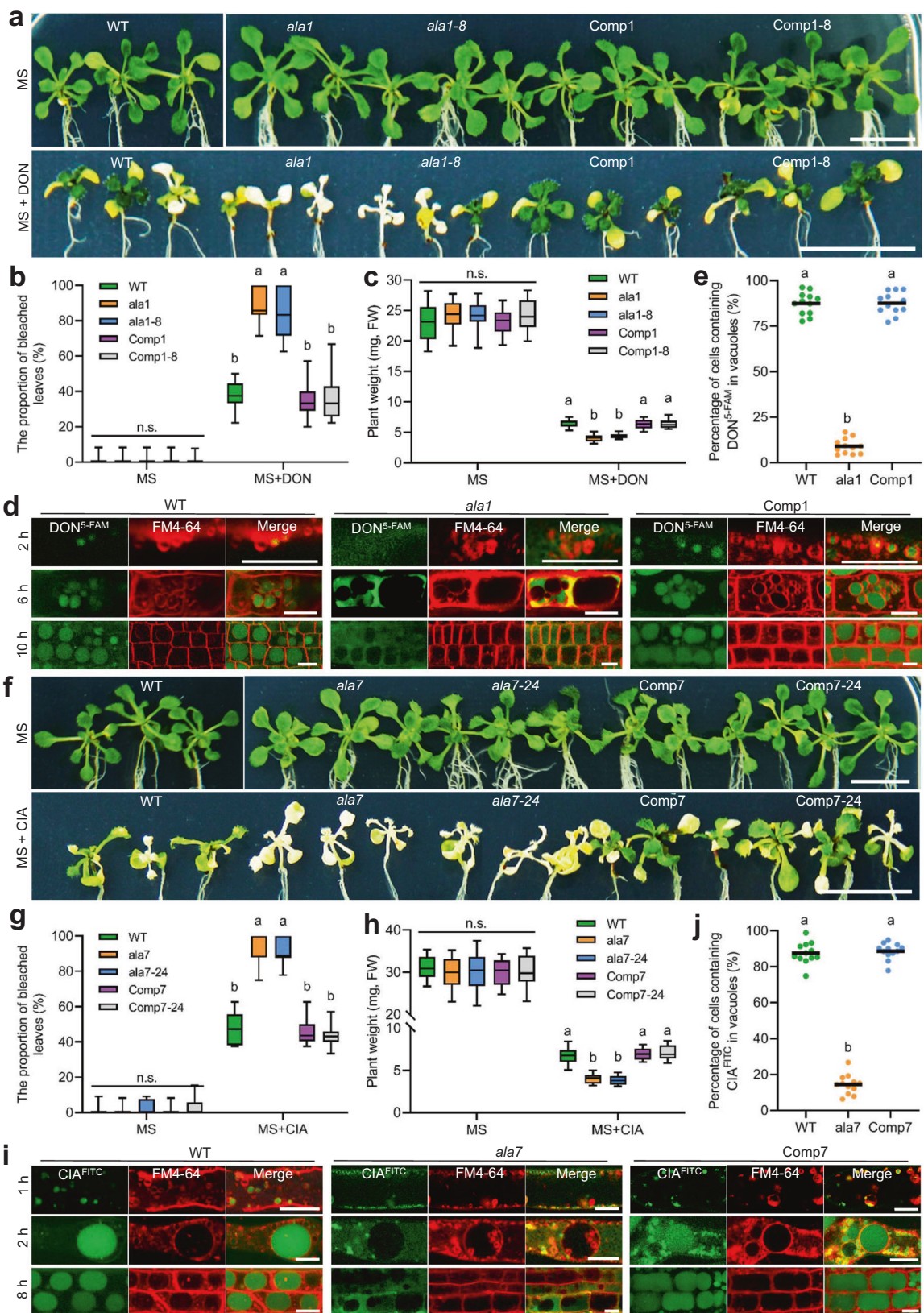

no CIA^FITC-containing vesicles were observed in root hair cells of the *ala7* mutant at 1 h and 2 h, whereas the signal clearly appeared in the vesicles at 2 h and accumulated in vacuoles of wild-type and complemented plants (*proAtALA7::AtALA7-eGFP*, Comp7) at 8 h (Fig. 1i, j). The decreased tolerance and defective

accumulation in vacuoles of DON/CIA were rescued by the expression of *AtALA1* and *AtALA7* in *ala1* and *ala7* plants, respectively (Fig. 1). These data reveal that in *Arabidopsis*, AtALA1 is responsible for the vacuole accumulation of DON and that AtALA7 is responsible for the vacuole accumulation of CIA.

**Fig. 1 AtALA1 is responsible for the vacuolar compartmentation of DON, and AtALA7 is responsible for that of CIA. a** DON tolerance assay of wild-type, and AtALA1 mutant (*ala1*, *ala1-8*) and complemented (Comp1, Comp1-8) *Arabidopsis* seedlings. The native promoter-controlled *AtALA1* was used to rescue the defect of AtALA1 in mutants. Six-day-old seedlings were transferred to MS media containing DON (50 µg/mL) for 7 d. WT, wild-type; *ala1* and *ala1-8*, AtALA1 loss-of-function mutant. Comp1 and Comp1-8, complemented *ala1* and *ala1-8* seedlings expressing *proAtALA1::EGFP-AtALA1*. DON, deoxynivalenol; *proAtALA1*, the native promoter of the *AtALA1* gene. Scale bar, 1 cm. **b** Proportion of bleached leaves suffering from DON. Data are presented as box-and-whisker plots of three replicates (eight plants each). **c** Weight of plants with or without DON treatment. Data are presented as box-and-whisker plots of three replicates (six plants each). **d** Distribution of DON[5-FAM] in root cells of wild-type, *ala1* mutant, and complemented *Arabidopsis* seedlings. Seedlings were treated with DON[5-FAM] (9 µg/mL) and FM4-64 (8 µM) for 2 h, 6 h, and 10 h. Confocal images at 2 h were collected from root hair cells, whereas those at 6 h and 10 h were collected from cells of the root meristem zone. At least three independent repeats were performed. DON[5-FAM], DON labeled with 5-carboxyfluorescein (5-FAM). Scale bar, 10 µm. **e** Percentage of cells containing DON[5-FAM] in their vacuoles at 10 h. Data are presented as dot plots ($n = 12$ roots). **f** CIA tolerance assay of wild-type, AtALA7 mutant (*ala7*, *ala7-24*) and complemented (Comp7, Comp7-24) *Arabidopsis* seedlings. The native promoter-controlled *AtALA7* was used to rescue the defect of AtALA7 in mutants. Six-day-old seedlings were transferred to MS media containing CIA (100 µg/mL) for 7 d. *ala7* and *ala7-24*, AtALA7 loss-of-function mutant. Comp7 and Comp7-24, complemented *ala7* and *ala7-24* mutants seedlings expressing *proAtALA7::AtALA7-eGFP*. CIA, cinnamate acetate; *proAtALA7*, the native promoter of the *AtALA7* gene. Scale bar, 1 cm. **g** Proportion of bleached leaves resulting from CIA treatment. Data are shown as box-and-whisker plots of three replicates (eight plants each). **h** Weight of plants with or without CIA treatment. Data are presented as box-and-whisker plots of three replicates (six plants each). **i** Distribution of CIA[FITC] in root cells of wild-type, *ala7* mutant and complemented (Comp7) *Arabidopsis* seedlings. The seedlings were treated with CIA[FITC] (5 µg/mL) and FM4-64 (8 µM) for 1 h, 2 h, and 8 h. Confocal images at 1 h and 2 h were collected from the root hair cells, while those at 10 h were collected from cells of the root meristem zone. At least three independent repeats were conducted. CIA[FITC], CIA labeled with fluorescein-isothiocyanate-isomer (FITC). Scale bar, 0.5 µm. **j** Percentage of cells containing CIA[FITC] in their vacuoles at 8 h. Data are presented as dot plots ($n = 12$ roots). Box-and-whisker plots show the medians (horizontal lines), upper and lower quartiles (box edges), and 1.5× the interquartile range (whiskers). Different letters in **b**, **c**, **e**, **g**, **h**, and **j** represent significant differences at $P < 0.05$ by one-way ANOVA with Tukey multiple comparisons test. n.s., not significant.

**DON/CIA are transported to vacuoles via AtALA1-/AtALA7-mediated vesicle transport**. To investigate the role of AtALA1 in DON transport, we generated transgenic *35S::RFP-AtALA1* and *proAtALA1::EGFP-AtALA1 Arabidopsis*. The DON[5-FAM] signal was colocalized with RFP-AtALA1 in the plasma membrane, vesicles and vacuoles of root hairs (Fig. 2a), showing that the accumulation of DON in vacuoles is associated with the vesicle-mediated transport pathway. To further trace the trafficking of DON in the cell, DON[5-FAM] treatment was performed separately in two transgenic *Arabidopsis* lines expressing RFP-fused RabA1d, an early endosome (EE) marker[80], and γ-Tip-RFP, a lytic vacuole marker[81]. The DON[5-FAM] signal appeared in RFP-RabA1d-indicated EE-like structures (Fig. 2b); the signal converged in vacuoles that were surrounded by γ-Tip-RFP (Fig. 2c). The subcellular localization of AtALA1 was observed in *proAtALA1::EGFP-AtALA1* transgenic *Arabidopsis*. In root hair cells, the signal was observed at the PM and tonoplast (Fig. 2d). The transient expression of AtALA1 in tobacco confirmed its localization at the PM (Supplementary Fig. 4a, b), which required the presence of ALIS1[82]. The colocalization analysis of EGFP-AtALA1 with the EE marker RFP-RabA1d and the late endosome (LE) marker RFP-RabF2a[83] showed that EGFP-AtALA1 was colocalized with the two marker proteins (Fig. 2e, f). These data demonstrated that AtALA1 was localized to the PM, EE, LE and tonoplasts.

To verify that the vacuole accumulation of DON[5-FAM] is associated with vesicle-mediated transport, wild-type seedlings were treated with the vesicle trafficking inhibitors tyrphostin A23 (TyrA23), brefeldin A (BFA), and wortmannin (Wm). TyrA23 is a well-characterized inhibitor of the clathrin-mediated endocytic pathway, which can specifically block the interaction between the tyrosine motif of the cargo and the adaptor protein AP-2[84,85]. Exposure of seedlings to TyrA23 led to a diffuse distribution of DON[5-FAM] in the cytoplasm rather than accumulation in mature vacuoles (Fig. 2g), indicating that endocytosis of DON may be clathrin-dependent. BFA is another vesicle trafficking inhibitor that affects the Golgi complex and Golgi retrograde transport, which usually results in the formation of the BFA compartment (BFA body) in plants[86,87]. In the BFA- and DON[5-FAM]- treated cells, massive DON[5-FAM], and BFA-induced FM4-64 aggregates formed, but the DON[5-FAM] signal was not colocalized to the BFA

bodies (Fig. 2g), confirming that the pathway for DON entering vacuoles is independent of the Golgi. Wm is a PI3 kinase-specific inhibitor, and this compound is widely used to observe vesicular trafficking routes in cells, including endocytosis and vesicle transport[88,89]. In Wm-treated cells, the DON[5-FAM] signal was diffusively distributed in the cytosol, and DON did not accumulate in vacuoles (Fig. 2g). Inhibition experiments and colocalization showed that DON was endocytosed from the PM and then transported through the EE and LE into lytic vacuoles, and this vesicle-associated transport might be mediated by AtALA1.

Similar approaches were used to detect the location and trafficking of AtALA7 and CIA in cells. The CIA[FITC] signal colocalized with AtALA7-RFP in vesicles in the root hairs of *35S::AtALA1-RFP* transgenic *Arabidopsis* (Fig. 3a). The endocytosis of CIA[FITC] in the cells of the root meristem region was slower than that in root hair cells (Fig. 1i). In the root meristem region, at 180 min and 183 min after exposure, CIA[FITC] fluorescent bubbles protruded from the PM; at 201 min, CIA[FITC] gathered together in the cytosol and formed prevacuolar compartment (PVC) structures (Fig. 3b); 6 hours later, the signal converged in lytic vacuoles (Fig. 3c). Inhibition assays (Fig. 3d) showed that the trafficking of CIA to the vacuole was also associated with vesicle-mediated transport. Subcellular localization results revealed that AtALA7-eGFP signals, similar to those of AtALA1, appeared in the PM (Fig. 3e), EE (Fig. 3f), LE (Fig. 3g), and tonoplasts (Fig. 3e). Transient expression of AtALA7-eGFP and ALIS1 in tobacco showed that similar to AtALA1, AtALA7 needs its β-subunit to leave the ER (Supplementary Fig. 4c, d). These data indicate a similar endocytosis pathway to transport CIA from the PM through the EE and LE into vacuoles.

**Overexpression of *AtALA7* promotes the accumulation of CIA in vacuoles and enhances resistance to *Verticillium* wilt disease**. To test the role of *AtALA7* in the tolerance of CIA, the gene was overexpressed under the control of the cauliflower mosaic virus 35S promoter (*35S::AtALA7*) in *Arabidopsis*. Two transgenic lines (OE-9 and OE-12) with high transcription levels of *AtALA7* (Supplementary Fig. 5b) were used for the bioassay of CIA resistance. Neither upregulation nor disruption of *AtALA7*

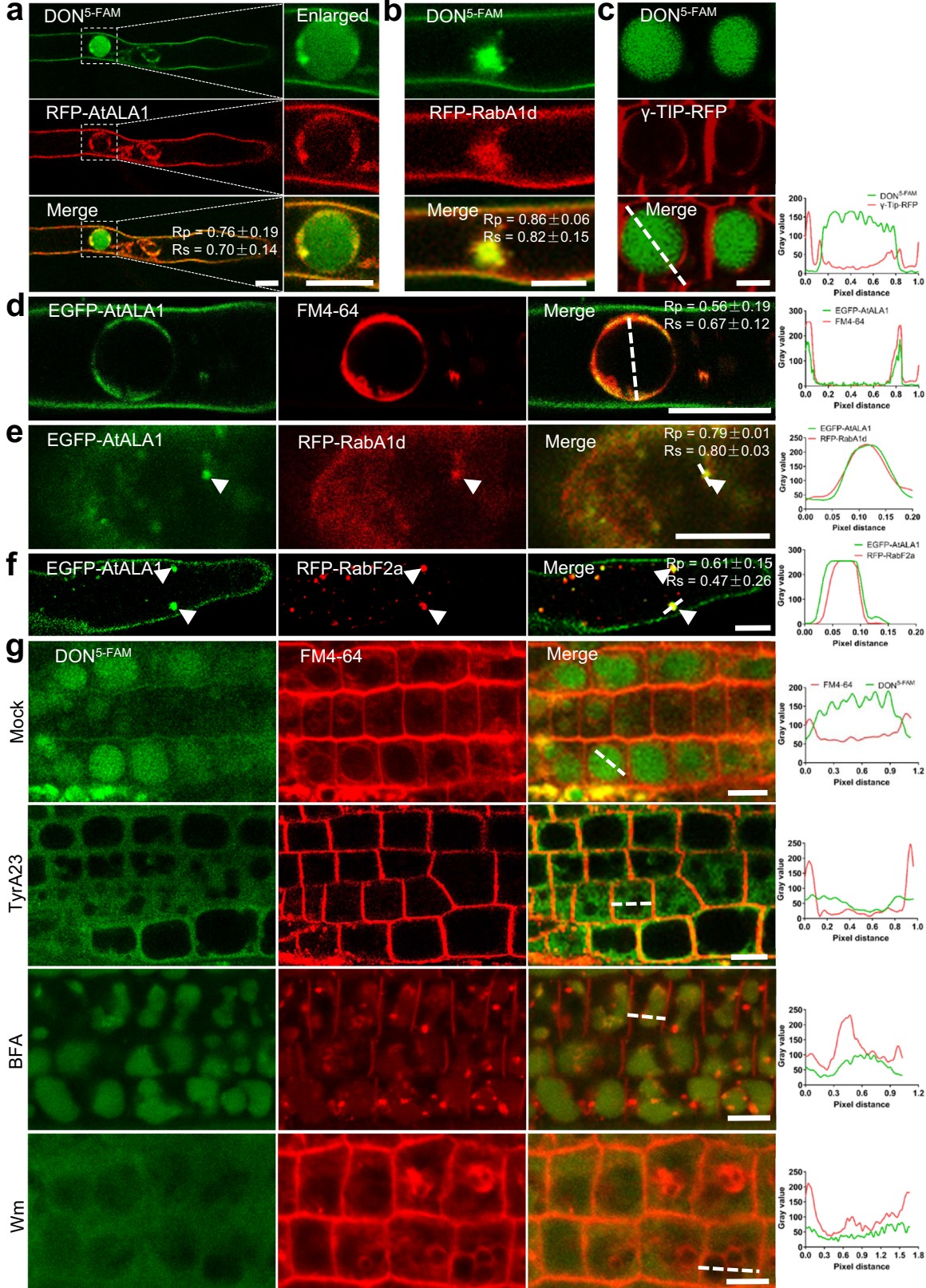

affected the growth of transgenic plants (Fig. 4a, b, Supplementary Fig. 1c and 5a, c, d). In the presence of CIA, wild-type plants showed toxic symptoms, including wilted leaves and decreased root growth. The symptoms were much more severe in the *ala7* and *ala7-24* mutants. In contrast, *AtALA7*-overexpressing lines (OE-9 and OE-12) showed less-severe disease than the wild-type

plants (Fig. 4a–c). We then observed the distribution of CIA[FITC] (5 μg/mL) in root cells and found that the CIA[FITC] signal in vacuoles of *35S::AtALA7* lines was much stronger than that of wild-type (Fig. 4d, e). To confirm this result, we generated *35S::AtALA7* tobacco and challenged two representative lines (Supplementary Fig. 6a) with CIA. The overexpression of *AtALA7*

**Fig. 2 DON[5-FAM] is transported via the EE and LE into lytic vacuoles. a** DON[5-FAM] colocalized with RFP-AtALA1 in vesicles and EE-like structures in root hairs. In the vacuole of the root hair, DON[5-FAM] colocalized with RFP-AtALA1 mainly in the tonoplasts. *35S::RFP-AtALA1* seedlings were treated with DON[5-FAM] (9 μg/mL) for 12 h. Scale bar, 5 μm. **b** DON[5-FAM] colocalized with RFP-RabA1d-labeled EE in root hairs. RFP-RabA1d, RFP-fused early endosome marker RabA1d. Six-day-old transgenic *35S::RFP-RabA1d* seedlings were treated with DON[5-FAM] (9 μg/mL) for 12 h. Scale bar, 5 μm. **c** DON[5-FAM] was transported into γ-Tip-RFP labeled lytic vacuoles in the cells at the root meristem region. Six-day-old *35S::γ-Tip-RFP* transgenic seedlings were treated with DON[5-FAM] (9 μg/mL) for 12 h. Scale bar, 5 μm. **d** In root hair cells of *proAtALA1::EGFP-AtALA1* transgenic *Arabidopsis*, EGFP-AtALA1 was localized to the PM and tonoplasts, as indicated by FM4-64 (8 μM) staining. Scale bar, 10 μm. **e** In root hair cells of transgenic *Arabidopsis* containing *proAtALA1::EGFP-AtALA1* and *35S::RFP-RabA1d*, EGFP-AtALA1 colocalized with the EE indicator RFP-RabA1d. Scale bar, 10 μm. **f** EGFP-AtALA1 colocalized with the LE indicator RFP-RabF2a in root hair cells of transgenic *Arabidopsis* containing *proAtALA1::EGFP-AtALA1* and *35S::RFP-RabF2a*. Scale bar, 10 μm. **g** Inhibition of TyrA23 (tyrphostin A23), BFA (brefeldin A), and Wm (wortmannin) on the transport of DON[5-FAM] to vacuoles. Mock, wild-type seedlings were treated with DON[5-FAM] (9 μg/mL) and FM4-64 (8 μM) for 12 h. TyrA23, BFA and Wm, Mock with TyrA23 (50 μM), BFA (50 μM) or Wm (33 μM) for 12 h. Scale bar, 5 μm. Experiments were independently repeated at least three times. The fluorescence density profile was measured along the dotted lines using ImageJ. The values of the Pearson correlation coefficient (Rp) and the Spearman correlation coefficient (Rs) show the extent of colocalization between the two target molecules. The values range between +1 (positive correlation) and −1 (negative correlation).

also increased the resistance of tobacco to CIA (Supplementary Fig. 6b, c).

We further tested the resistance of AtALA7 transgenic *Arabidopsis* to a strong pathogenic strain of *V. dahliae* L2-1. After inoculation, typical symptoms of Verticillium wilt appeared in the wild-type control, whereas severe symptoms were observed in the *ala7* and *ala7-24* mutants; in contrast, *AtALA7*-overexpressing lines showed less-severe disease (Fig. 4f, g). The relative fungal biomass of *V. dahliae* in the *35S::AtALA7* line (OE-9 and OE-12) was lower than that of the wild-type and mutant plants (Fig. 4h). Likewise, *35S::AtALA7* tobacco also displayed a significantly lower disease index than the wild-type (Fig. 4i, j). These results suggest that the overexpression of *AtALA7* can increase plant resistance against different *V. dahliae* strains (e.g., L2-1 and V991).

**Overexpression of *AtALA1* promotes DON transport to vacuoles and increases the resistance of *Arabidopsis* to DON and *F. graminearum*.** To confirm the role of *AtALA1* in DON detoxification, the gene was overexpressed in *Arabidopsis*. Similar to *AtALA7*, neither upregulation nor knockout of *AtALA1* significantly affected the growth of the plants (Fig. 5a, b, and Supplementary Fig. 5e, g, h). Transgenic lines growing on the DON-containing medium exhibited better plant growth with more biomass, while those of *ala1* and *ala1-8* mutants displayed poorer growth and less biomass relative to the wild-type control (Fig. 5a–d). In addition, the *35S::AtALA1* lines exhibited a stronger ability to transport DON[5-FAM] into vacuoles than wild-type (Fig. 5e, f).

The DON-producing pathogen *F. graminearum* was then used to test disease resistance *via* the *Fusarium-Arabidopsis* floral pathosystem[90]. Six days after inoculation with *F. graminearum* spores, mycelia emerged on the surface of buds. More mycelia were observed on the buds of mutant plants (*ala1* and *ala1-8*) compared with those of the wild-type, while the *35S::AtALA1 Arabidopsis* (OE-15 and OE-25) buds had remarkably fewer mycelia (Fig. 5g). Nine days after inoculation, severe disease symptoms of *F. graminearum*, such as brown tissue discoloration and aerial mycelium on the surface of siliques, were observed on mutant plants (*ala1* and *ala1-8*) and wild-type, whereas the disease symptoms of *35S::AtALA1* lines were mild (Fig. 5g–i). The *F. graminearum* abundance in *ala1* mutant plants was significantly higher than that in wild-type; in contrast, this abundance was much lower in *35S::AtALA1* lines (Fig. 5j). Determination of the DON concentration in siliques harvested from these infected plants showed that the content of *AtALA1* lines OE-15 and OE-25 was 1.50 ± 0.25 ng/g and 1.25 ± 0.29 ng/g, respectively, which was significantly lower than that in the wild-type siliques (3.53 ± 0.75 ng/g; Fig. 5k). However, the concentrations in the

*ala1* and *ala1-8* siliques (6.19 ± 0.56 ng/g and 7.22 ± 0.54 ng/g) were largely higher than that in the wild-type (Fig. 5k). These data demonstrate the significant efficacy of AtALA1 in DON detoxification.

**Heterologous overexpression of *AtALA1* enhances the resistance of maize to ear rot and significantly reduces the DON concentration in seeds.** To test the DON detoxification ability of AtALA1 in monocot crops, we expressed *AtALA1* under the control of the ubiquitin promoter in maize. Three transgenic *Ubi1::AtALA1* lines (# 4, # 13, and # 15) with relatively high levels of *AtALA1* transcription were used in this study (Supplementary Fig. 7a, b). The transgenic maize plants were indistinguishable from the wild-type in plant growth and seed size (Supplementary Fig. 7c–h and Supplementary Table 2). DON ranging from 30 to 90 μM inhibited radicle elongation in a dose-dependent manner (Supplementary Fig. 6d, e); thus, we treated seeds of transgenic maize with 60 μM DON to investigate this resistance. Although the radicle elongation of *Ubi1::AtALA1* maize seedlings (T₃) was inhibited by DON, the extent of inhibition was much weaker than that of the wild-type (Fig. 6a, b). The DON[5-FAM] accumulation assay showed that a large amount of fluorescent signal was compartmentalized in vacuoles of the *Ubi1::AtALA1* lines (# 4), which was much stronger than that in the vacuoles of the wild-type (Fig. 6c, d).

Next, *F. graminearum* was inoculated into the coleoptiles of 6-day-old seedlings. Six days later, a large area of wild-type coleoptiles became dark brown, and aerial mycelia were visible on the surface. In contrast, the disease severity on *Ubi1::AtALA1* coleoptiles was strikingly lower than that on the control (Fig. 6e, f). Then, spores of *F. graminearum* (1.3 × 10^8 spores/mL) were inoculated on young silks by spraying. Forty days later, the top part (nearly one-third) of the wild-type cobs became dark brown, and some cobs even began to rot. White mycelia appeared not only on the top part of the cob but also on the middle part. In contrast, in *Ubi1::AtALA1* transgenic maize (# 4, # 13, and # 15), the disease symptom was observed just at the very top region of the cobs, and no mycelia were found on the ears (Fig. 6g). Statistically, the disease incidence rate and disease area were significantly decreased in *Ubi1::AtALA1* transgenic plants compared with the wild-type (Fig. 6h, i). Notably, the DON concentration was also decreased in transgenic maize grains. For example, in lines # 4 and # 15, the DON contents were 62.6 ± 19.8 ng/g and 51.8 ± 2.8 ng/g, respectively, which were significantly lower than those in wild-type grains (122.3 ± 15.3 ng/g; Fig. 6j). These data showed that heterologous expression of *AtALA1* significantly increased the resistance of maize to *F. graminearum* and dramatically decreased the DON content in seeds.

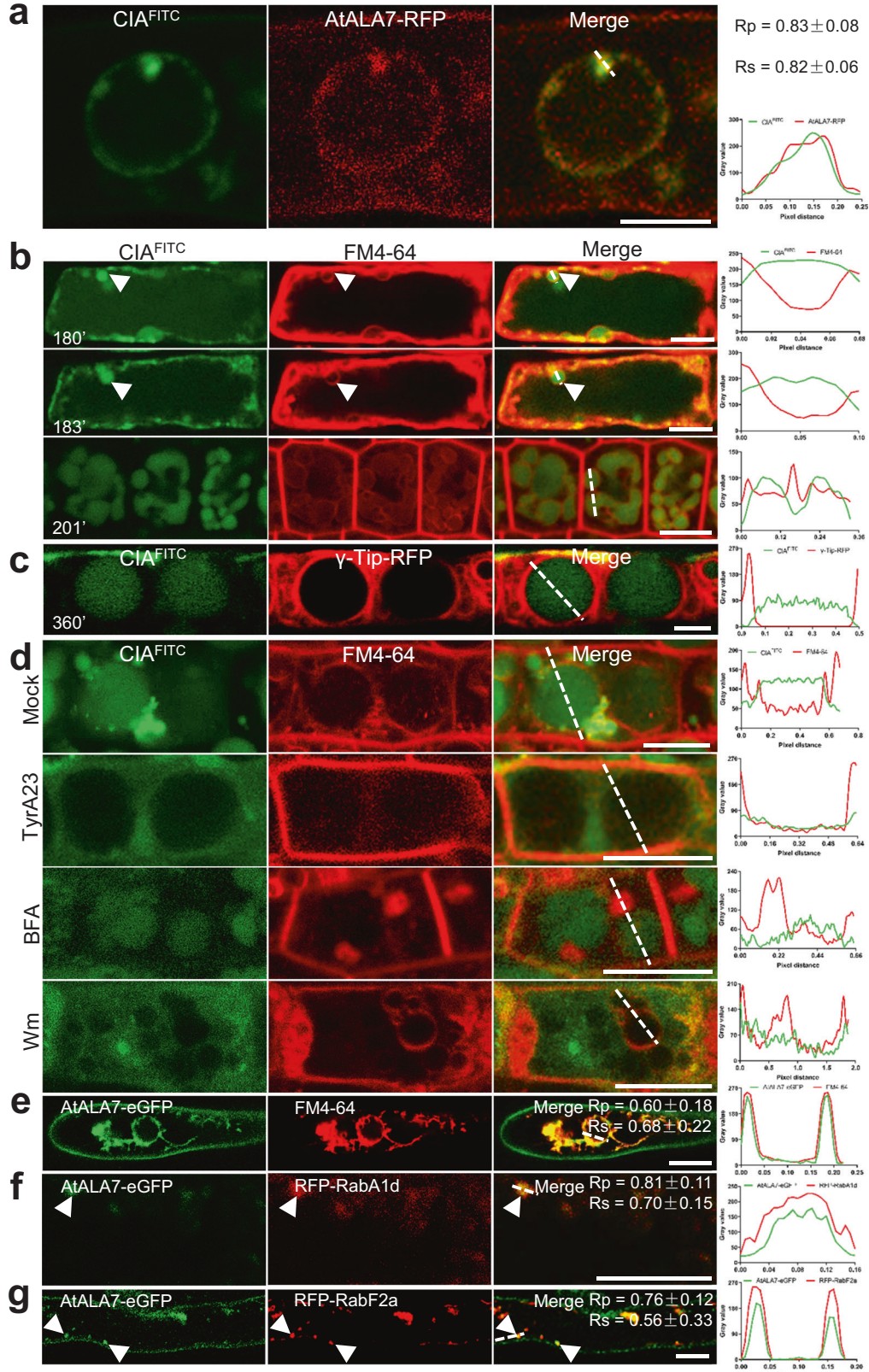

## Discussion

Based on their infection and feeding strategy, plant pathogens can be divided into three major groups: biotrophs, necrotrophs, and hemibiotrophs. Biotrophic pathogens colonize living tissues without killing their hosts, whereas necrotrophic pathogens kill host cells to obtain nutrients from dead tissues; hemibiotrophic pathogens start infection as a biotroph and then live as sapro-phytes on dead tissues[2,91,92]. When plants are infected by bio-trophic pathogens, hypersensitive responses (HR), including oxidative bursts, are induced to trigger programmed cell death (PCD), which in turn confines the infection to dead cells. In contrast, necrotrophic fungi usually produce toxins to subvert

**Fig. 3 CIA^FITC is transported via the EE and LE into lytic vacuoles. a** CIA^FITC colocalized with AtALA7-RFP in vesicles and vacuoles of the root hair. *35S::AtALA7-RFP* seedlings were treated with CIA^FITC (15 µg/mL) for 6 h. Scale bar, 5 µm. **b** CIA^FITC and FM4-64 fluorescence in root cells of wild-type *Arabidopsis* at 180 min, 183 min, and 201 min after treatment with CIA^FITC (15 µg/mL) and FM4-64 (8 µM). Scale bar, 10 µm. The triangle indicates the location of CIA^FITC aggregates. **c** Distribution of CIA^FITC in root cells of transgenic *Arabidopsis 35S::γ-Tip-RFP*. Seedlings were treated with CIA^FITC (5 µg/mL) for 6 h; CIA^FITC fluorescence mainly accumulated in lytic vacuoles. Scale bar, 10 µm. **d** Inhibition of TyrA23, BFA, and Wm on the accumulation of CIA^FITC in vacuoles of root cells. *Arabidopsis* wild-type seedlings were treated with 5 µg/mL CIA^FITC and 8 µM FM4-64 (Mock), and extra TyrA23 (50 µM), BFA (50 µM) or Wm (33 µM) for 6 h. Scale bar, 10 µm. **e** AtALA7-eGFP was localized to the PM and tonoplasts in root hair cells of *proAtALA7::AtALA7-eGFP* transgenic *Arabidopsis*, which were stained with FM4-64 (8 µM). *proAtALA7*, the native promoter of *AtALA7*. Scale bar, 10 µm. **f** AtALA7-eGFP colocalized with the EE indicator RFP-RabA1d in root hair cells of transgenic *Arabidopsis* containing *proAtALA7::AtALA7-eGFP* and *35S::RFP-RabA1d*. Scale bar, 10 µm. **g** AtALA7-eGFP colocalized with the LE marker RFP-RabF2a in root hair cells of transgenic *Arabidopsis* containing *proAtALA7::AtALA7-eGFP* and *35S::RFP-RabF2a*. Scale bar, 10 µm. Experiments were independently repeated at least three times. The fluorescence density profiles were measured along the dotted lines using ImageJ. The values of the Pearson correlation coefficient (Rp) and the Spearman correlation coefficient (Rs) show the extent of colocalization between the two target molecules. The values range between +1 (positive correlation) and −1 (negative correlation).

host defenses, such as HR, to accelerate the PCD of host cells, thus benefiting their colonization in plants[2]. Pathogenesis is more complicated for hemibiotrophic pathogens: they can generate low levels of toxins to suppress PCD, thus facilitating pathogen colonization and infection during the initial biotrophic phase[93]. They can also produce high levels of toxins to stimulate cell death, thus releasing nutrients to facilitate necrotrophic fungal growth during the necrotrophic phase[35,94,95]. The complicated pathogenic process makes it difficult to control hemibiotrophic pathogens. As toxins play a key role in both biotrophic and necrotrophic pathogenicity, detoxification is, therefore, a logical strategy for controlling these diseases.

Although the development of resistance to FHB by chemical modification/inactivation of trichothecenes have been achieved in wheat and barley[42], this strategy has apparent limitations. One drawback of this strategy is the potential undesirable side effects on plant growth or development, resulting from the manipulation of metabolism. For example, glycosylation mediated by glycosyltransferase has been utilized in plants for DON detoxification[36,96]. Since some glycosyltransferases have a wide range of substrates, including plant hormones, their overexpression in plants may interfere with hormone homeostasis, thus resulting in abnormal growth[36,41,97]. For instance, although the overexpression of *Arabidopsis* glycosyltransferase DOGT1, which converts toxic DON to nontoxic D3G, increased the resistance of *Arabidopsis* to DON, transgenic *Arabidopsis* displayed a brassinosteroid deficiency-like dwarf phenotype due to the formation of brassinosteroid glucosides[36,41]. More importantly, glycosylated *Fusarium* toxins can be reversed in animals during digestion[42]. In fact, all modified forms of *Fusarium* toxins likely have the same toxicity to human and animal health as their parent compounds[43]. In vesicle-mediated detoxification, in contrast, toxins are wrapped into vesicles surrounded by membranes and then transported into vacuoles where the toxins are compartmentalized and degraded; thus, there is no chance for the toxins to be in contact with their targets. In our experiments, no significant alterations in plant growth or development were observed in the transgenic plants overexpressing *AtALA1/AtALA7*. Most importantly, the resistance to *F. graminearum* and *V. dahliae* is significantly increased, and the DON contamination in maize grains is largely reduced.

Our data show that the entry speed of CIA into the cytosol is faster than that of DON (Supplementary Fig. 3e, f). Because of its lipophilic character and small molecular weight (176.21 Da), CIA can cross the plasma membrane and enter the cell (Figs. 1i, 3b and Supplementary Fig. 2g, 3e), resulting in wilt symptoms in plants[98] (Fig. 1f, 4a). In contrast, DON has a polar molecule structure. The ability of DON to enter cells by diffusing across the cell membrane, therefore, is very low[99]. The possible way of DON enters cells may be through an unknown membrane-associated

passive transporter or endocytosis/pinocytosis process[99]. Once entering cells, the amphiphilic nature of DON can increase its cytotoxicity on the cell by interactions with a number of targets, including ribosomes[100] and mitochondria[101,102]. Our study shows that either lipophilic CIA or amphiphilic DON can be wrapped in vesicles and transported from the PM to vacuoles; thus, the amount of toxins entering cells through other pathways is significantly reduced. The detoxification mechanism mediated by AtALA1/AtALA7 is shown in Fig. 7.

In the present study, using DON and CIA as probes, we successfully identified the *Arabidopsis* AtALA1 and AtALA7 responsible for the detoxification of mycotoxins in plants. We, therefore, hypothesize that cargoes transported by P4 ATPase-associated vesicle trafficking for detoxification should be a group of molecules with similar physicochemical or biological properties. AtALA1 may be responsible for the transport of some amphiphilic molecules, such as DON, whereas AtALA7 may be responsible for the transport of lipophilic compounds, such as CIA. If this is the case, P4 ATPase(s) can potentially be utilized for detoxifying different toxins in various species. Details about what properties the compounds should have and how the P4 ATPases recognize their cargos await further investigation. In addition to AtALA1 and AtALA7, other ALAs may also participate in the detoxification of other toxins. The method described here allows us to identify more P4 ATPases that are responsible for transporting specific molecules.

In brief, our findings may help drive research on increasing plant resistance against intractable plant diseases caused by toxin-producing pathogens, such as rice sheath blight, rice blast, FHB, and Verticillium wilt. This P4 ATPase-conferred resistance can potentially be achieved in different species against different diseases, and the detoxification strategy is also applicable to cope with mycotoxin contamination in food and feed.

## Methods

**Plant materials and growth conditions**. The *Arabidopsis thaliana* T-DNA insertion mutants *ala1* (salk_056947), *ala1-8* (salk_002106), *ala2* (salk_070975), *ala3* (salk_082157), *ala4* (salk_078875), *ala5* (salk_056611), *ala6* (salk_150173), *ala7* (salk_125598), *ala7-24* (salk_063917), *ala8* (salk_071661), *ala9* (salk_070924), *ala10* (salk_107029) and *ala11* (salk_056947) were obtained from the SALK collection. These mutants were genotyped using a PCR-based approach (http://signal.salk.edu/tdnaprimers.2.html). The PCR primer sequences are given in Supplementary Table 1.

Seeds of *Arabidopsis thaliana* (ecotype Columbia, Col-0) were surface-sterilized in 75% alcohol for 20 min, rinsed 3~5 times with sterile water, and sown on MS medium (Murashige and Skoog medium, M519, Phytotech) containing 1.5% (w/v) sucrose and 1% (w/v) gelrite. Plants were incubated at 4 °C for 2 d in the dark and then transferred to a growth chamber at 21 ± 2 °C with a 16 h light/8 h dark photoperiod.

Tobacco was grown in a greenhouse at 25 °C under an 18 h light/6 h dark photoperiod. *Agrobacterium tumefaciens* strain EHA105 was used to transform *Nicotiana tabacum cv.* Xanthi[103]. The maize cultivar Hi II[104] was used for genetic transformation and DON inhibition assays. Seeds were incubated in sterile water for 24 h in the dark and placed between two sheets of sterile Whatman filter paper

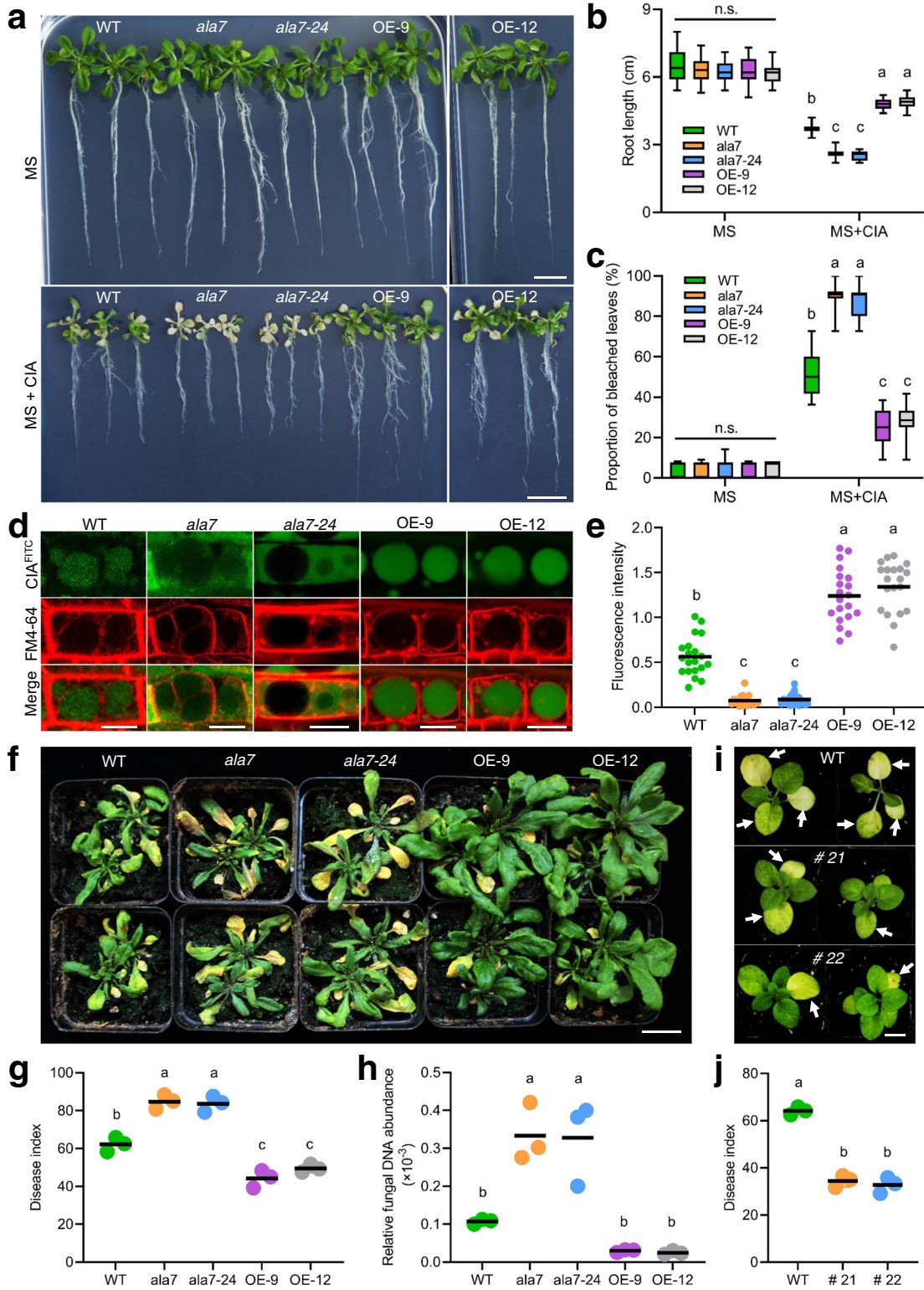

with 5 mL of sterile water inside a Petri dish. Seeds were incubated for 5 d in total darkness at 25°C to ensure germination.

**Gene cloning and transgenic plant production**. cDNA fragments were cloned into a pLGN vector modified from pCambia2300[105]. In the T-DNA region of pLGN, there is a neomycin phosphotransferase II (*NPTII*) selection marker fused with a *GUS* reporter gene under the control of a *35S* promoter. For the *35S::AtALA1* constructs, the coding sequence of *AtALA1* was cloned from

*Arabidopsis* (Col-0) using the primer pair AtALA1-F tail/-R (*Sal*I) tail and digested with *Sma*I and *Sal*I. The resulting fragment was then inserted into the pLGN vector to form pLGN-*35S::AtALA1*. For the *35S::AtALA7* constructs, the coding sequence of *AtALA7* was cloned from *Arabidopsis* (Col-0) using the primer pair AtALA7-F (*Spe*I) tail/-R (*Kpn*I) tail and digested with *Spe*I and *Kpn*I. The resulting fragment was inserted into the pLGN vector to form pLGN-*35S::AtALA7*. The coding sequence of enhanced green fluorescent protein (EGFP) was amplified from plasmid eGFP-C1 (6084-1, Clontech). To fuse the N-terminus of AtALA1 with EGFP together, EGFP and AtALA1 were cloned with the primer pairs EGFP-F

**Fig. 4 Overexpression of *AtALA7* promotes the transport of CIA to vacuoles and enhances the resistance of *Arabidopsis* and tobacco to CIA and *V. dahliae*. a** In the presence of CIA, the growth of wild-type seedlings was significantly inhibited, and the inhibition was more severe in the *ala7* mutant; in contrast, 35S::*AtALA7* seedlings showed resistance to the toxin. Seedlings were grown on MS media containing CIA (100 μg/mL) for 9 d. WT, wild-type; *ala7* and *ala7-24*, AtALA7 loss-of-function mutant; OE-9 and OE-12, 35S::*AtALA7* lines. Scale bar, 1 cm. **b** Root length of the wild-type, *ala7* mutant and *AtALA7*-overexpressing seedlings with or without CIA treatment. Data are presented as box-and-whisker plots of three replicates (nine plants each). **c** Proportion of bleached leaves suffering from CIA treatment. Data are presented as box-and-whisker plots of three replicates (nine plants each). **d** Distribution of CIA$^{FITC}$ in root cells of wild-type, mutant (*ala7*, *ala7-24*), and *AtALA7*-overexpressing lines (OE-9, OE-12). Scale bar, 10 μm. **e** Fluorescence intensity per vacuole in root cells of wild-type, mutant (*ala7*, *ala7-24*) and 35S::*AtALA7*-overexpressing plants (OE-9). Data are shown as dot plots ($n = 20$ vacuoles). Fluorescence intensity was estimated by ImageJ. **f** Verticillium wilt symptoms of wild-type, mutant (*ala7*, *ala7-24*), and *AtALA7*-overexpressing *Arabidopsis* lines (OE-9, OE-12). Plants were inoculated with *V. dahliae* strain L2-1 (3 mL, $2 \times 10^8$ spores/mL) for 21 d. Scale bar, 2 cm. **g** Disease index of wild-type, mutant (*ala7*, *ala7-24*), and *AtALA7*-overexpressing lines (OE-9 and OE-12). Data are shown as dot plots of three replicates (30 plants each). **h** Relative fungal biomass of *V. dahliae* shown as dot plots of three replicates (eight plants per group). **i** Verticillium wilt symptoms of wild-type and *AtALA7* transgenic tobacco lines (# 21 and # 22). Seedlings were inoculated with *V. dahliae* strain V991 (10 mL, $2 \times 10^8$ spores/mL) and photographed at 3 weeks. The white arrow indicates diseased leaves. Scale bar, 2 cm. **j** Disease index of wild-type and *AtALA7* transgenic tobacco plants. Data are presented as dot plots of three replicates (30 plants each). Box-and-whisker plots show the medians (horizontal lines), upper and lower quartiles (box edges), and 1.5× the interquartile range (whiskers). Different letters in **b**, **c**, **e**, **g**, **h**, and **j** represent significant differences at $P < 0.05$ by one-way ANOVA with Tukey multiple comparisons test. n.s., not significant.

tail/-R (+linker) tail and AtALA1-F (+linker) tail/-R (*Sal*I) tail, respectively, and the resulting products were fused by overlap extension PCR to link EGFP with AtALA1 (EGFP-AtALA1). Then, the PCR product was digested with *Sma*I/*Sal*I and inserted into the pLGN vector to produce the pLGN-*35S::EGFP-AtALA1* cassette. For the *proAtALA1::EGFP-AtALA1* construct, the promoter sequence of *AtALA1* (2039 bp) was cloned and fused with EGFP-AtALA1 by overlap PCR. The resulting fragment was inserted into the pLGN vector (*Bam*HI/*Hind*III) via homologous recombinase (Boer, MC001M-50 Rxns). To link the C-terminus of AtALA7 with EGFP, EGFP, and AtALA7 were cloned with primer pairs AtALA7-F (*Spe*I)/-R (+linker) tails and EGFP-F (+linker) tail/-R (*Kpn*I) tail, respectively, and the resulting products were fused by overlap extension PCR to link AtALA7 and EGFP (AtALA7-eGFP). Then, the PCR product was digested with *Spe*I/*Kpn*I and inserted into the pLGN vector to form the pLGN-*35S::AtALA7-eGFP* cassette. For the *proAtALA7::AtALA7-eGFP* construct, the promoter sequence of *AtALA7* (2053 bp) was cloned and fused with AtALA7-eGFP by overlap PCR. The resulting fragment was inserted into the pLGN vector (*Bam*HI/*Hind*III) via homologous recombinase (Boer, MC001M-50 Rxns). For organelle marker protein construction, the CDS of Pip2a (At3g53420), RabA1d (At4g18800), RabF2a (At5g45130), and γ-Tip (At2g36830) were fused with a linker and RFP by overlap PCR. The resulting fragments were inserted into a digested pLGN vector (*Bam*HI/*Eco*RI for Pip2a/γ-Tip and *Bam*HI/*Kpn*I for RabA1d/RabF2a) via homologous recombinase. For the *Ubi1::AtALA1* construction, the coding sequence of *AtALA1* was cloned with primer pairs AtALA1-F (+Ubi1, *Stu*I) tail/-R (+Ubi1, *Kpn*I) tail, and the PCR product was digested with *Stu*I/*Kpn*. The resulting fragment was inserted into a modified pLGN vector (35S was replaced by the ubiquitin promoter Ubi1). PCR primer sequences are shown in Supplementary Table 1.

Maize transformation was conducted using particle bombardment[106]. *Arabidopsis* transformation was performed by the floral-dip method[107] using *Agrobacterium tumefaciens* strain GV3101. Homozygous transgenic plants were screened by kanamycin (50 μg/mL) and PCR.

**RNA extraction and quantitative (real-time) reverse transcription PCR (qRT-PCR)**. RNA was extracted using the EASY Spin Plant RNA Kit (Aidlab, China) according to the manufacturer's instructions. Approximately 1.0 μg of RNA was used for cDNA synthesis with a RevertAid First Strand cDNA Synthesis Kit (Fermentas, Canada). A 10-μL reaction mixture containing cDNA and SYBR Green Super-mix (Bio-Rad) was used for qRT-PCR following the manufacturer's instructions. The thermal cycling program consisted of an initial denaturation at 95 °C for 3 min, followed by 40 cycles of 95 °C for 15 s, 58 °C for 20 s and 72 °C for 30 s. Data were analyzed by CFX Manager 3.1 software (Bio-Rad, USA). Each test was repeated at least three times. *AtActin2*, *ZmEF1α*, and *NtActin* were used as reference genes in *Arabidopsis*, maize, and tobacco, respectively. The primers used for qRT-PCR are listed in Supplementary Table 1.

**Subcellular localization**. The colocalization of AtALA1 and AtALA7 with different organelle markers fused with RFP was carried out by transient expression in tobacco epidermal cells or by the floral-dip method in *Arabidopsis* with *Agrobacterium tumefaciens* strain GV3101 containing *EGFP-AtALA1* (or *AtALA7-eGFP*) and organelle markers (PiP2a-RFP, RFP-RabF2a, RFP-HDEL, and RFP-RabA1d). For transient expression in tobacco epidermal cells, *A. tumefaciens* cells were pelleted by centrifugation and resuspended in infiltration solution (10 mM MgCl₂, 10 mM MES and 0.2 mM acetosyringone) until the OD$_{600}$ reached 0.5; this was followed by incubation in the dark for 3 h before co-infiltration into leaves of *N. benthamiana*. The fluorescent signal was detected with a microscope (Leica SP8, Germany) at 48-72 h after co-infiltration. For the localization of AtALA1 (or

AtALA7) in PM and tonoplasts, roots of *EGFP::AtALA1* transgenic *Arabidopsis* were stained with FM4-64 (8 μM, F34653, Thermo Scientific). The colocalization of AtALA1 (or AtALA7) to the EE was observed in root hair cells of transgenic *Arabidopsis* containing EGFP-AtALA1 (or AtALA7-eGFP) and an EE marker (RFP-RabA1d). The colocalization of AtALA1 (or AtALA7) to the LE was observed in root hair cells of transgenic *Arabidopsis* containing EGFP-AtALA1 (or AtALA7-eGFP) and an LE marker (RFP-RabF2a).

**Pathogen inoculation and disease scoring**. The wild-type *F. graminearum* strain PH-1 (3.4584, CGMCC) was cultured on potato dextrose agar (PDA, 254920, BD-Difco) plates at 26 °C[108]. Falcate conidia were collected from PDA plates, transferred into liquid mung bean (*Vigna radiata*) medium[109], and cultivated at 26 °C for 3 d with shaking at 200 rpm. The conidial suspension was filtered through four-layer gauze and centrifuged at $5000 \times g$ for 10 min. The harvested conidia were resuspended in sterile water. The suspension was used for inoculation within 2 h.

*Arabidopsis* plants with both open flowers on the terminal inflorescence and two to three developing siliques were used for inoculation assays[90]. In brief, the *F. graminearum* strain PH-1[108] spores suspension ($1.3 \times 10^8$ spores/mL) was spray-inoculated into the inflorescence until droplet run-off occurred from the inflorescence. Control plants were inoculated in the same way using deionized water. The inoculated plants were then kept in large plastic propagators at 100% relative humidity for the next 6 days. For the first 2 days, the chambers were shaded with capillary matting to exclude light. Disease scoring was assessed as described previously[90] with slight modification. In brief, three separate floral subcomponents, namely, flowers (F) that were either open flowers or closed buds at inoculation, new siliques (NS) that were fully open flowers at inoculation, and older siliques (OS) already present at inoculation, were considered. An increasing numerical score was used to quantify the abundance of aerial mycelium on a tissue surface (0, 1), as well as the increased severity of the disease symptoms visible on plant tissue as the invasion process progressed (3-7). Intermediate scores of 2 and 4 (F) and 2, 4, and 6 (NS and OS) were reserved for when all the tissue on a single plant exhibited the disease phenotype described for the preceding score. The final *Fusarium–Arabidopsis* disease (FAD) value was calculated by the addition of the three subcomponent scores, FAD = F + NS + OS.

For maize ear inoculation, 3-day-old silk ears of maize plants were spray-inoculated with fresh spores suspension (2 mL, $5 \times 10^5$ spores/mL) of *F. graminearum* strain PH-1. After inoculation, the maize ears were covered with black bags and kept at 95% humidity for 48 h. The disease was rated by the percentage of infected seeds at 40 dpi. For maize coleoptile inoculation, 6-day-old seedlings were used for the coleoptile infection assay according to Zhang et al.[109] with modifications. In brief, 6 days after seed sowing, the top 2–3 mm of the coleoptiles were removed, and the seedlings were drop-inoculated with 8 μL of fresh spores suspension ($7.5 \times 10^7$ spores/mL) of *F. graminearum* strain PH-1 for 6 d. Mock-inoculated coleoptiles (with water) served as controls. After inoculation, the inoculated maize seedlings were grown in a growth chamber at 25 °C and 95% humidity for 48 h.

For *V. dahliae* inoculation, *Arabidopsis* and tobacco seedlings were inoculated with *V. dahliae* strains L2-1 and V991, respectively. The inoculation procedure was as follows: at least 50 seedlings of each construct were dipped into a spore suspension liquid containing $2 \times 10^8$ spores/mL. The disease index (DI) of Verticillium wilt was scored using at least 50 plants per treatment. The severity of disease symptoms was classified as level 0-4. Here, 0 means no visible wilting or chlorosis symptoms[110]. 1 means 0-25% (inclusive) of true leaves wilted, chlorosis or dropped off; 2 means 25-50% (inclusive) of true leaves wilted, chlorosis or dropped off; 3 means 50-75% (inclusive) of true leaves wilted, chlorosis or dropped off; and 4 means 75-100% (inclusive) of true leaves wilted, chlorosis or dropped off.

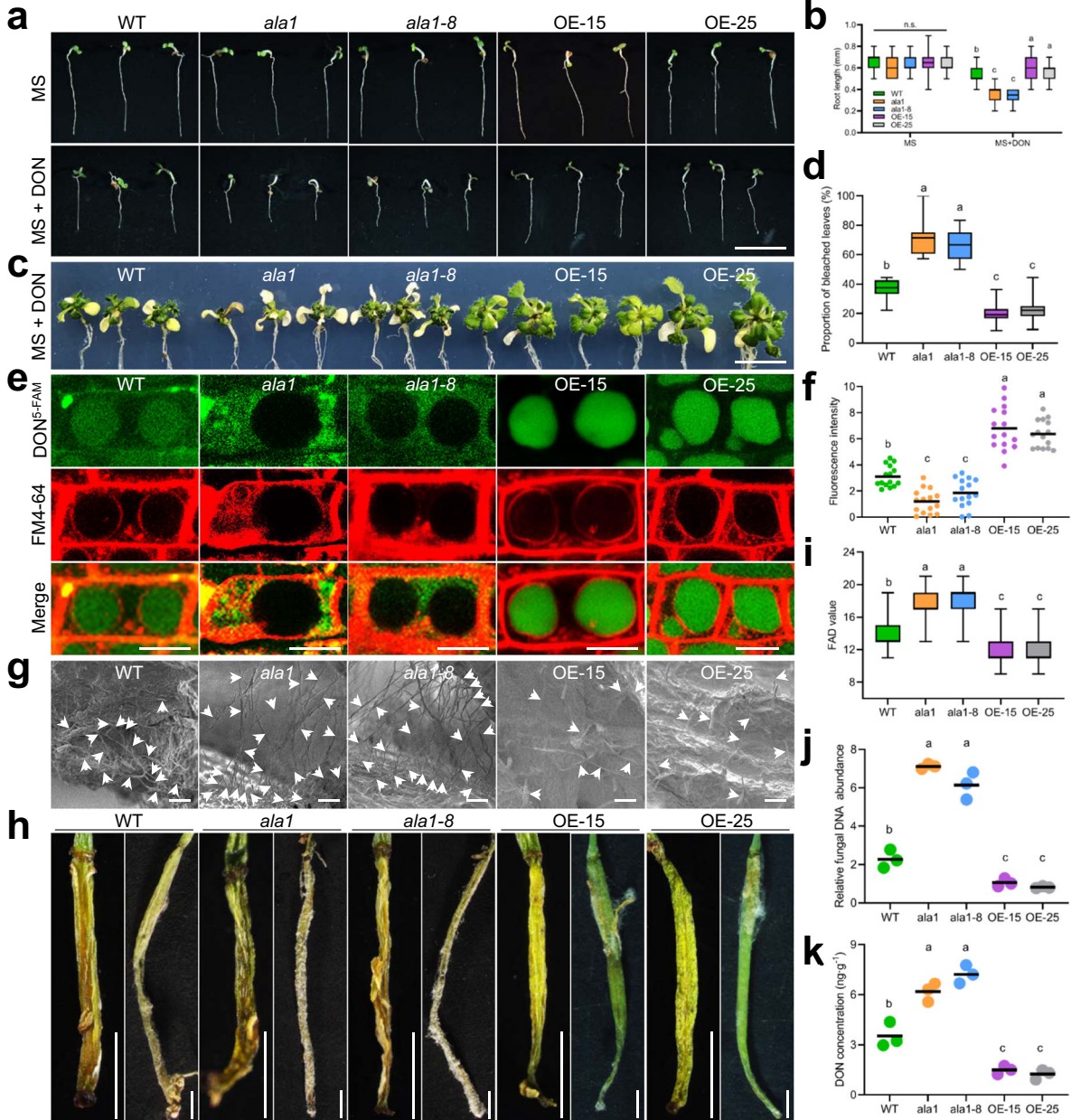

**Fig. 5 Overexpression of *AtALA1* promotes the transport of DON to vacuoles and increases the resistance of *Arabidopsis* to DON and *F. graminearum*.**
**a** DON resistance assay of wild-type, mutant (*ala1*, *ala1-8*), and *35S::AtALA1*-overexpressing lines (OE-15 and OE-25). Two-day-old seedlings were treated with DON (1 μg/mL) for 5 d. WT, wild-type *Arabidopsis*; *ala1* and *ala1-8*, AtALA1 loss-of-function mutant; OE-15 and OE-25, transgenic *35S::AtALA1* *Arabidopsis* lines. Scale bar, 5 mm. **b** Root length of plants with or without DON treatment. The results are shown with box-and-whisker plots of three replicates (12 plants each). **c** DON resistance assay of wild-type, mutant (*ala1*, *ala1-8*), and *35S::AtALA1*-overexpressing lines (OE-15 and OE-25). Six-day-old seedlings were transferred to MS media containing DON (50 μg/mL) for 12 d. Scale bar, 20 μm. **d** Proportion of bleached leaves resulting from DON treatment. Data are presented as box-and-whisker plots of three replicates (eight plants each). **e** Distribution of DON[5-FAM] in root cells of wild-type, mutant (*ala1*, *ala1-8*), and *AtALA1*-overexpressing lines (OE-15, OE-25). Six-day-old seedlings were treated with DON[5-FAM] (9 μg/mL) and FM4-64 (8 μM) for 12 h. Scale bar, 10 μm. **f** Fluorescence intensity of DON[5-FAM] in root cells of wild-type, mutant (*ala1*, *ala1-8*), and *AtALA1*-overexpressing lines (OE-15, OE-25). The results are shown with dot plots (*n* = 15 vacuoles). **g** *F. graminearum* mycelia on the surface of buds at 6 dpi (days post-inoculation). Plants were spray-inoculated with *F. graminearum* (1.3 × 10⁸ spores/mL). Arrows indicate the hyphae or spores of *F. graminearum*. Scale bar, 20 μm. The experiment was repeated three times independently with similar results. **h** Symptoms of *F. graminearum* on siliques (9 dpi) of wild-type, mutant (*ala1*, *ala1-8*), and *AtALA1*-overexpressing lines (OE-15, OE-25). Plants were spray-inoculated with *F. graminearum* (1.3 × 10⁸ spores/mL). Scale bar, 1 mm. **i** Disease severity estimated by the FAD (*Fusarium–Arabidopsis* disease) value[90]. The results are shown with box-and-whisker plots of three replicates (nine plants each). **j** Real-time PCR analysis of the DNA abundance of *F. graminearum* in flowers at 9 dpi. Data are presented as dot plots of three replicates (12 plants each). The copy number of the *F. graminearum* TRI6 gene was estimated relative to that of the *Arabidopsis* actin2 gene. **k** DON concentration from *F. graminearum*-infected siliques of wild-type, mutant (*ala1*, *ala1-8*), and *AtALA1*-overexpressing lines (OE-15, OE-25). The results are shown as dot plots of three replicates (five plants each). DON concentration was determined by LC-MS/MS. Box-and-whisker plots show the medians (horizontal lines), upper and lower quartiles (box edges), and 1.5× the interquartile range (whiskers). Different letters in **b**, **d**, **f**, **i**–**k** represent significant differences at *P* < 0.05 by one-way ANOVA with Tukey multiple comparisons test. n.s., not significant.

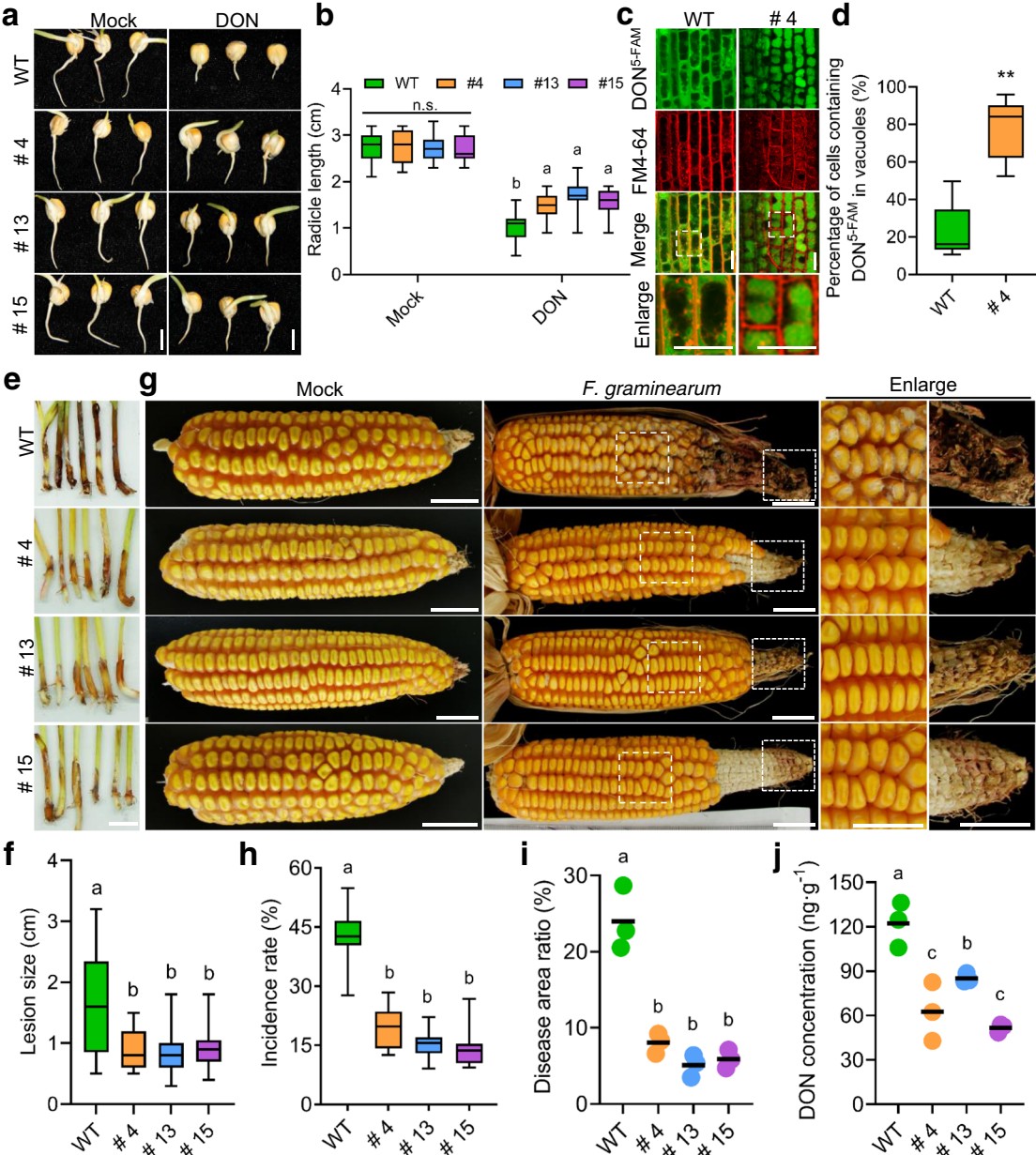

**Fig. 6 Heterologous overexpression of *AtALA1* in maize enhances resistance to *F. graminearum* and decreases DON contamination in seeds. a** DON inhibition assay of *Ubi1*::*AtALA1* transgenic lines and wild-type seeds. Seeds were soaked with DON (60 μM) or sterilized water and then germinated in moisture chambers for 2 d. WT, wild-type maize (cultivar Hi II); # 4, # 13 and # 15, transgenic *Ubi1*::*AtALA1* seeds (T₃). Scale bar, 0.5 cm. **b** Radicle length of *Ubi1*::*AtALA1* transgenic lines and wild-type seeds in the absence or presence of DON (60 μM). The results are shown with box-and-whisker plots of three replicates (5 roots each). **c** Distribution of DON⁵⁻ᶠᴬᴹ in root cells of *Ubi1*::*AtALA1* transgenic lines and wild-type. Three-day-old seedlings were treated with DON⁵⁻ᶠᴬᴹ (30 μg/mL) and FM4-64 (20 μM) for 12 h. Scale bar, 20 μm. **d** Statistics of the proportion of DON⁵⁻ᶠᴬᴹ-containing vacuoles in the root cells, as shown with box-and-whisker plots of three replicates (nine roots each). \*\**p* < 0.01, significantly different with respect to wild-type according to Student's *t* test (two-tailed). **e** Stalk rot symptoms on coleoptiles of *Ubi1*::*AtALA1* transgenic lines and wild type. *F. graminearum* spores suspension (8 μL, 7.5 × 10⁷ spores/mL) was dropped onto the coleoptiles of 6-day-old seedlings. To promote infection, the coleoptiles were wounded with a scalpel. Scale bar, 1 cm. **f** Statistics for lesion length on the *F. graminearum*-infected coleoptiles (6 dpi) of *Ubi1*::*AtALA1* transgenic lines and wild-type maize. Six-day-old seedlings were treated with *F. graminearum* (7.5 × 10⁷ spores/mL). Data are presented as box-and-whisker plots of three replicates (eight plants each). g Rot symptoms of ears (40 dpi) of *Ubi1*::*AtALA1* transgenic lines and wild-type maize. Maize ears (three days after pollination) of *Ubi1*::*AtALA1* transgenic lines and wild-type were inoculated with *F. graminearum* (5 × 10⁵ spores/mL). Mock, maize ears (three days after pollination) of *Ubi1*::*AtALA1* transgenic lines and wild-type plants were inoculated with deionized water as a control. Scale bar, 2 cm. h Statistics for the incidence rate of *F. graminearum*-infected maize ears. Data are presented as box-and-whisker plots of three replicates (five maize ears each). **i** Statistics for the diseased area ratio of *F. graminearum*-infected maize ears, shown as dot plots of three replicates (four ears each from individual plants). **j** DON concentration in the seeds from *F. graminearum*-infected corncobs of *Ubi1*::*AtALA1* transgenic lines and wild-type maize. DON concentration was determined by LC-MS/MS. The results are shown as dot plots of three replicates (each replicate had 80 g seeds from four ears, which were harvested from individual plants). Box-and-whisker plots show the medians (horizontal lines), upper and lower quartiles (box edges), and 1.5× the interquartile range (whiskers). Different letters in **b**, **f**, and **h**–**j** represent significant differences at *P* < 0.05 by one-way ANOVA with Tukey multiple comparisons test. n.s., not significant.

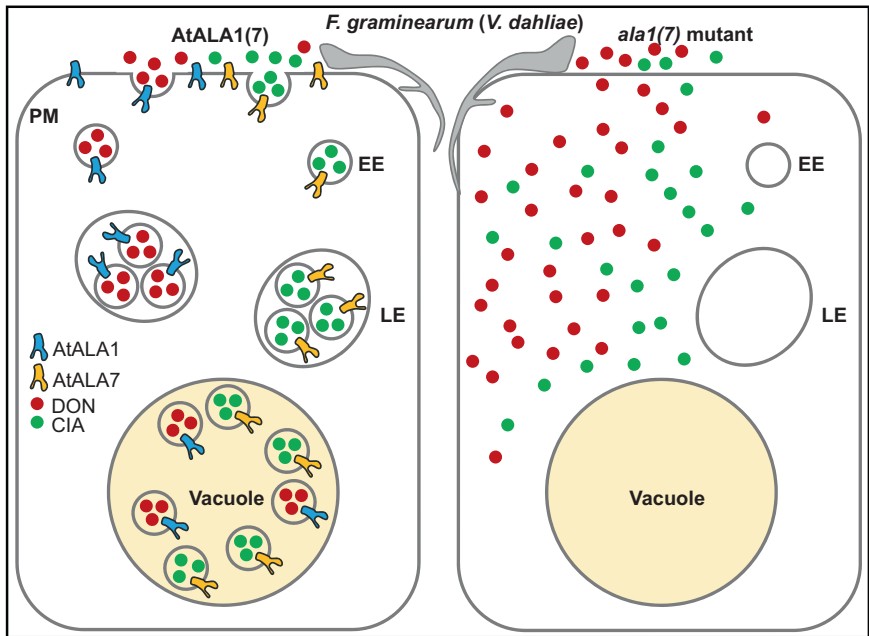

**Fig. 7 A model showing *Arabidopsis* AtALA1-/AtALA7-mediated vesicle trafficking of DON/CIA for mycotoxin detoxification.** Toxins (DON or CIA) are wrapped into vesicles on the PM, and the formation of these structures is mediated by P4 ATPases (AtALA1/AtALA7). Then, the toxins are transported into vacuoles, where they are compartmented and degraded. Lipophilic CIA is capable of crossing the plasma membrane barrier into cells and produces wilt symptoms. With an unknown membrane-associated passive transporter or endocytosis/pinocytosis mechanism[99], DON can enter the cell, resulting in cytotoxicity due to its interactions with a number of targets[100–102]. By avoiding contact with the targets, the vesicle/vacuolar compartmentation of the toxins protects the plant innate immune system from damage, thus allowing plants to maintain their innate immunity against the invasion of pathogens and consequently increasing resistance to mycotoxin-associated diseases, such as Fusarium head blight (FHB) and Verticillium wilt. *PM* plasma membrane; *EE* early endosome, *LE* late endosome; *DON* deoxynivalenol, *CIA* cinnamyl acetate.

The DI was calculated according to the following formula: DI = [∑ (disease grades × number of infected plants)/ (total checked plants × 4)] × 100.

**DON and CIA tolerance assays**. For the DON and CIA treatments in *Arabidopsis*, 6-day-old plants were transferred to MS solid medium containing DON (50 μg/mL, APEXBIO) and CIA (100 μg/mL, Sigma-Aldrich) for 7 d or 9 d, respectively. For DON treatment in maize, germinated seeds with radicle lengths of ~ 0.5 cm were transferred to sterile water containing DON (60 μM) for 2 d.

**Quantification of DON concentration by LC-MS/MS**. Eighty grams of maize seeds and two grams of *Arabidopsis* siliques infected with *F. graminearum* were used for DON content analysis. The frozen tissue samples were ground to a fine powder in the presence of liquid nitrogen. Four volumes of methanol-water (80:20, v/v) were added to the powder, and the mixture was then shaken for 3 h at room temperature. After centrifugation at 5000 *g* for 10 min, the supernatant was transferred to an Eppendorf tube and concentrated to dry by vacuum. The concentrate was resuspended in methanol-water (50:50, v/v) and filtered through a 0.22 μm pore filter membrane. Forty microlitres of the dissolved samples were used for DON quantification using a 4000 Q-Trap LC-ESI-MS/MS system (SCIEX, USA) with a C18 column (2.1 × 150 mm, 3.5 μm, Agilent) at 30 °C[111]. A series of gradients of methanol/0.1% formic acid (v/v) was applied at a flow rate of 0.165 mL/min as follows: 0~10 min, 22% to 40%; 10~15 min, 40~100%; 15~25 min, 100%; 25~40 min, 22%. DON was detected in the positive ion mode. Each test was repeated with three biological replicates.

**Inhibitor treatments and FM4-64 staining**. DON was labeled with 5-carboxyfluorescein (5-FAM, HY-66022, APEXBIO). CIA was labeled with fluorescein-isothiocyanate (FITC, HY-66019, MCE). The labeled compounds were produced by Fanbo Biochemicals Co. Ltd. (Haidian HighTech Business Park, Beijing, China).

For the CIA^FITC endocytosis inhibition assay, wild-type *Arabidopsis* seedlings were incubated in MS medium (Murashige and Skoog medium, M519, Phytotech) containing CIA^FITC (5 μg/mL), FM4-64 (8 μM, F34653, Thermo Scientific), and extra TyrA23 (50 μM, APEXBIO), BFA (50 μM, Klamar) or Wm (33 μM, Solarbio) for 6 h. For the DON^5-FAM endocytosis inhibition assay, wild-type *Arabidopsis* seedlings were treated with DON^5-FAM (9 μg/mL) and FM4-64 (8 μM) for 12 h (Mock), or mock with TyrA23 (50 μM), BFA (50 μM) or Wm (33 μM). The samples were washed with deionized water three times before imaging.

**Scanning electron microscopy (SEM)**. For the observation of mycelium and conidia of *F. graminearum* on the surface of *Arabidopsis* buds, the buds were frozen in liquid nitrogen, transferred into the cryostorage of a preparation chamber for platinum coating, and then imaged using an S-3400N SEM (Hitachi) at an accelerating voltage of 2 kV.

**Living cell imaging**. Living cell images were mainly acquired with a Leica SP8 point-scanning confocal system (Leica Microsystems, Germany). Images of the root meristem zone were acquired with a ×40 objective, and root hair cells were acquired with a ×63 objective. The pinhole aperture was 1. EGFP, CIA^FITC, and DON^5-FAM (green channel) were excited at 488 nm, and the emission spectra were between 495 and 540 nm. FM4-64 and RFP (red channel) were excited at 552 nm, and the emission spectra were between 570 and 610 nm. The scan speed was 400 Hz at a resolution of 1024 × 1024 pixels. All fluorescent signals were quantified using ImageJ.

For the observation of *F. graminearum* mycelium on *Arabidopsis* siliques, the SZX-ILLB2-200 stereomicroscope imaging system (Olympus) was used.

**Statistical analysis**. Statistical analyses were performed with Student's *t* test or one-way ANOVA with a Tukey multiple comparisons test. Each experiment was repeated at least three times. The fluorescence intensity, Rp, and Rs were calculated using the image processing program ImageJ (http://rsbweb.nih.gov/ij/). Standard errors and standard deviations were calculated using Excel (Microsoft, version 16) and SPSS (IBM, version 19).

**Reporting summary**. Further information on research design is available in the Nature Research Reporting Summary linked to this article.

## Data availability

The LC-MS/MS data have been deposited in Metabolights with the data set identifier MTBLS3522. Data supporting the major findings of this work are available within the paper and its supplementary files. All additional data are available from the corresponding author upon request. Source data are provided with this paper.

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

## Acknowledgements

This work was supported by the Chinese Ministry of Science and Technology of China (Grant 2016YFD0100505 to Y.P.), National Transgenic New Species Breeding Major Project of China (2016ZX08005-003-004 to Y.P.), Chongqing Research Program of Basic Research and Frontier Technology (cstc2017jcyjB0316 to X.L.), and the Graduate Student Research Innovation Project of Chongqing (CYB17072 to F.W.). We thank Dr. Caixia Gao (Institute of Genetics and Developmental Biology, Chinese Academy of Sciences) for her help in maize transformation, professor Zhiying Ma (Hebei Agricultural University, China), and professor Guiliang Jian (The Institute of Plant Protection, Chinese Academy of Agricultural Sciences) for their kind donation of *V. dahliae* strain L2-1 and V991. We are grateful to professor Zhengqiang Ma (Nanjing Agricultural University, Nanjing, China) and Dr. Dapeng Zhang (USDA ARS, BARC, PSI) for their critical reading of the manuscript.

## Author contributions

Y.P. conceived and supervised the project. F.W. performed ALAs' identification and subcellular localization, plant transformation, assays of disease resistance and toxin resistance, plant management, and data analysis; Y.L. and H.R. participated in P4 ATPases identification and protein subcellular localization; X.L. J.H., M.S., J.Z., Y.C., H.L., J.Z., and S.S. participated in plant transformation and plant management; L.H., J.Z, Y.F., X.Y., D.J., and M.Z. participated in toxin determination, confocal and SEM observations, and data processing. Y.P. and F.W. wrote the paper with input from all other authors.

## Competing interests

The authors declare no competing interests.
