## [Peer Review File · Nature Communications]

Arabidopsis P4 ATPase-mediated cell detoxification confers resistance to *Fusarium graminearum* and *Verticillium dahliae*REVIEWER COMMENTS

Reviewer #1 (Remarks to the Author):

Wang et al. in the manuscript entitled “Arabidopsis P4 ATPase-mediated cell detoxification confers the resistance to *Fusarium graminearum* and *Verticillium dahliae*” argue that ALA1 and ALA7 detoxify fungal toxins, DON and CIA, by trafficking them into vacuoles. All data are likely to support the authors’ claim, but more precise experimental results should be presented to be published in the journal. In addition, the authors are likely to pick or select only a part of data to support their argument.

1. To show cellular localization of ALA1 and ALA7, the authors generated transgenic Arabidopsis plants expressing GFP or RFP-fused proteins by 35S promoter. Using these plants or sometimes by heterologous expression in tobacco with some plant organellar markers, the authors argue that ALA1/7 are transported from the PM via endosomes to vacuole. However, as the authors may know, by overexpression proteins can be localized to additional cellular compartments to their original sites. Therefore, the authors should generate and use plants expressing fusion proteins by native promoters to precisely observe their localization in plant cells.
2. The authors found that DON or CIA are not accumulated in FM4-64-stained vacuoles in *ala1* or *ala7* mutant in Fig. 1. But, DON and CIA induces leaf whitening in plants, which is accelerated in the mutant plants, as shown in the same Fig. 1. This may mean that cell death progress is more rapidly induced in those mutants by DON and CIA. This additionally suggest that no more vacuolar localization of DON and CIA might not simply result from the absence of ALA1/7 but from cell death-associated breaking of cellular compartmentalization. Therefore, the authors have to show the localization of DON and CIA at earlier time points as Fig. 3b, or by movie in mutant plants together with WT.
3. To test disease resistance to fungi, the authors generated transgenic Arabidopsis plants overexpressing ALA1 or ALA7 only. I wonder whether transgenic plant overexpressing fluorescently tagged proteins also show similar enhanced resistance to tested fungi? Are there any specific reasons to use ALA1 or ALA7 only to generate transgenic plants for resistance test?
4. The authors show leaf whitening (resistance to DON and CIA), root length (ALA7 overexpression) or plant weight (ALA1 overexpression) depending on an experiment. Are there any reasons for this? The authors should show all three kinds of results regardless of different experimental approaches.
5. In many figures, arrowheads are not clearly indicated.
6. Pictures in Fig. 2b and c were obtained at the same time point after DON treatment. Why DON is distinctly localized? In addition, why localization patterns in Fig. 2e are not observed in Fig. 2d?

Reviewer #2 (Remarks to the Author):

The manuscript presents the putative role of two P4 ATPases (ALA1 and ALA7) in detoxification of fungal toxins. P4 ATPases are lipid translocators involved in vesicle formation, lipid signaling and generation of an asymmetry lipid distribution that can be used to recruit proteins to the surface of the membrane. Based on KO mutants, protein overexpression and the use of secretory pathway inhibitors, the authors demonstrate that overexpression of arabidopsis ALA proteins can be used to increase the resistance of several plant species to fungal pathogens. While their findings are interesting, the mechanism behind the observed phenotype are not clear and several points require attention.

General comments

1. The English language relatively good, but there are many paragraphs where the grammatical constructions get too complicated or the use of the language is not correct. E.g. "This strategy enables hosts to be prevented from the toxicity of the mycotoxins" or "Recently, it was turned out that..." - The authors should consider professional language editing.
2. While I am not a statistics expert, I am not sure the statistical analysis is correct for most of the figures. In most figures, the authors present a Student's t-test, which is meant to compare two means. However, most experiments include three or more means, as there is always a control and at least two mutant/complemented lines. I believe the correct way to do this analysis is using a test of the type of ANOVA, which compares several different means, even if the authors are only interested in showing the difference between each mutant and the wild type. This type of analysis has been used for Sup. Fig. 2, for instance.
3. In general, it is difficult to assess the quality of the bioimaging work, such as the subcellular localizations and the secretory pathway inhibitor experiments, as no mention is made of number of repetitions and no quantifications are provided in most cases. The authors should clearly state how many repetitions of each experiment were made and provide quantifications and statistical analyses.

Introduction

1. "The P4 subfamily of P-type ATPases (P4 ATPases) functions as phospholipid flippases that translocate specific phospholipid substrates from the exoplasmic/luminal leaflet to the cytoplasmic leaflet of biological membranes,..."

All plant P4 ATPases characterized to date and most P4 ATPases from other organisms are phospholipid transporters, but several P4 ATPases from humans and yeast have now been shown to transport lipids not containing a phosphate group, so please correct his sentence to account for that fact.

2. When describing the general characteristics and physiological function of P4 ATPases in different organisms the authors cite a number of reviews published from 1996 to 2016 (refs. 52-60), several of which are too old to reflect our knowledge of the proteins. In the past 4-5 years the field has made huge advances and several updated reviews have been published, including some specifically devoted to plant

P4 ATPases. While some of the references could stay, at least the oldest one should be updated. In addition, the authors should avoid using original citations for the general subjects or they should otherwise be fair and cite all the original papers on the subject. Some suggestions for the authors:

López-Marqués, R.L. et al. (2021) Dynamic membranes: the multiple roles of P4 and P5 ATPases. *Plant Physiol.* DOI: 10.1093/plphys/kiab065

Shin, H.-W. and Takatsu, H. (2019) Substrates of P4-ATPases: beyond aminophospholipids (phosphatidylserine and phosphatidylethanolamine). *FASEB J.* 33, 3087–3096

Best, J.T. et al. (2019) Phospholipid flippases in membrane remodeling and transport carrier biogenesis. *Curr. Opin. Cell Biol.* 59, 8–15

Nintemann, S.J. et al. (2019) Catch you on the flip side: A critical review of flippase mutant phenotypes. *Trends Plant Sci.* 24, 468–478

Andersen, J.P. et al. (2016) P4-ATPases as phospholipid flippases—Structure, function, and enigmas. *Front. Physiol.* 7, 275

3. When citing the role of plant P4 ATPases in plant development, one reference is missing:

Davis, J.A. et al. (2020) The lipid flippases ALA4 and ALA5 play critical roles in cell expansion and plant growth. *Plant Physiol.* 182, 2111–2125

4. In the list of physiological roles of plant P4 ATPases "...temperature tolerance⁶¹⁻⁶³, disease resistance⁶⁴⁻⁶⁶, auxin polar distribution⁶⁷, plant growth and development,...", I suggest that the authors also include plant fertility, as several P4 ATPases are important for this function

McDowell, S.C. et al. (2013) Loss of the *Arabidopsis thaliana* P4-ATPase ALA3 reduces adaptability to temperature stresses and impairs vegetative, pollen, and ovule development. *PLoS One* 8, e62577

McDowell, S.C. et al. (2015) Loss of the *Arabidopsis thaliana* P4-ATPases ALA6 and ALA7 impairs pollen fitness and alters the pollen tube plasma membrane. *Front. Plant Sci.* 6, 197

Zhou, Y. et al. (2020) The tip-localized phosphatidylserine established by *Arabidopsis* ALA3 is crucial for Rab GTPase-mediated vesicle trafficking and pollen tube growth. *Plant Cell* 32, 3170–3187

Results

1. The authors show that *ala1* and *ala7* mutants have defective accumulation of fungal toxins in their vacuoles and that this phenotype can be complemented by the expression of *AtALA1* and *AtALA7*, respectively. Therefore the authors conclude: "These data reveal that, in *Arabidopsis*, *AtALA1* is responsible for the DON vacuole accumulation and *AtALA7* is for the CIA vacuole accumulation".

In order to make this conclusion, the authors are lacking at least one additional KO line for each ALA that demonstrates that the phenotype is directly related to the ALA protein. While complementation shows that the ALA proteins can complement the phenotypes observed, the authors are overexpressing the proteins all over the plants using a 35S promoter. Thus, even if the phenotypes are not directly related to the deletion of ALA1 or ALA7 but to another gene deleted in the background, the ALA proteins might have an activity that can, when overexpressed, rescue the phenotype.

2. "DON5-FAM signal appeared in RFP-RabA1d-indicated EE-like structures (Fig. 2b); the signal was finally converged in vacuoles which were surrounded by the γ -Tip (Fig. 2c)."

The authors do not make a time-course of the accumulation of fluorescent DON, so they should not make any statement that refers to this. In this case, "finally" implies that they are showing the substance travelling through EE to the vacuole with time, which is not the case.

3. "In Arabidopsis root cells, EGFP-AtALA1 localized at the PM and tonoplast, which was stained by FM4-64 (Fig. 2d,e)"

The localization of the protein with the tonoplast is not at all clear. The authors present a picture in which FM4-64, which clearly colocalizes with the tonoplast does not really overlap with the signal for ALA1. ALA1 seems to be present in vesicular structures surrounding the vacuole (perhaps prevacuolar compartments) and a diffuse membrane surrounding the vacuole (ER?), but there is no clear tonoplast colocalization. This is supported by the low Rp value.

4. "In tobacco epidermal cells, EGFP-AtALA1 was colocalized with RFP-RabF2a (Fig. 2g)."

It has been shown before that ALA1 does not leave the ER in tobacco epidermal cells in the absence of a co-expressed beta subunit (López-Marqués, R.L. et al. (2012) A putative plant aminophospholipid flippase, the Arabidopsis P4 ATPase ALA1, localizes to the plasma membrane following association with a β -subunit. PLoS One 7, e33042). When co-expressed with a beta subunit, ALA1 travels to the plasma membrane and no association with endomembranes was observed in these experiments. How can the authors explain this discrepancy?

5. "In the BFA and DON5-FAM treated cells, massive aggregates of DON5-FAM and BFA induced FM4-64 aggregates were formed, but the DON5-FAM signal was not colocalized to the BFA bodies (Fig. 2h)."

Can the authors rule out that what they call "massive aggregates" are not simply an accumulation of DON in the vacuoles? Trafficking from the Golgi is affected by BFA, but the authors have not shown so far that the Golgi is involved in trafficking of DON to the vacuole. In fact, the signals obtained for BFA treatment for CIA-treated plants are clearly different than those obtained for DON, which suggests the two substances do not follow the same route to the vacuole.

6. The subcellular localization of ALA7 with GFP (Figure 3e) is not convincing. The ALA7 signal looks like the ER membrane, which is attached to the plasma membrane through contact points even after plasmolysis. To date, no ALA protein has been shown to be able to leave the ER in the absence of a beta subunit. To make these experiments credible (both for ALA1 and ALA7), the tobacco localization needs to be repeated in the presence of a beta subunit.

7. Do *ala1* or *ala7* plants show any cellular phenotypes in vesicular trafficking that can support their role? All the experiments are based on protein overexpression and there is no direct evidence of the involvement of any of the two ALA proteins in vesicle formation under native conditions.

Discussion

1. “With amphiphilic structure, in the contrary, the ability of DON to enter the cells through diffusing across ...”

Amphiphilic means “having an affinity for both acid and basic dyes.” Do the authors mean “amphiphilic” (having both hydrophilic and hydrophobic parts)? The term is used repeatedly in the text.

2. “Possibly, AtALA1 is responsible for transport of some amphiphilic molecules like DON; while AtALA7 is responsible for that of lipophilic compounds as CIA.”

The authors have not provided evidence that their results can be generalized like this to other molecules, and should tone down this statement.

3. The authors seem to have forgotten to write a reference to Sup. Fig. 7 in the text. In this figure, they present a model of how ALA proteins might be involved in the observed detoxification effects. As mentioned in my comments above, the BFA treatment of plants suggests slightly different transport pathways to the vacuole for both substances and a putative model should reflect these differences.

4. The role of ALAs in detoxification of fungal toxins has not yet been described, but a role on detoxification of heavy metals was found for ALA4 (Sanz-Fernández, M. et al. (2017) Screening Arabidopsis mutants in genes useful for phytoremediation. *J. Hazard. Mater.* 335, 143–151). The authors could use this as an argument for the general use of flippases in detoxification.

Figures

Sup. Figure 2: For the treatment with CIA there is not quantification of the amount of cells containing fluorescent CIA. Please include this information. The co-localization of signals along an axis is not giving any more information than what can be easily seen in the pictures and does not provide any information on whether the pictures are representative of a population.

Sup. Figure 3: Panel f, 6 hours. The image is not really showing a vacuolar localization of the fluorescent DON and it looks very much like bleed through signal.

Figure 2, panel d, Wm- The image for the fluorescent CIA is very poor. The authors should consider showing an image that does not look like background signal and clearly shows the fluorescence is inside the cells, if it is present there.

Methods

1. In “Plant Materials and Growth Conditions” please indicate the growth conditions for tobacco plants as well.

2. In the generation of transgenic plants, please include the procedure for tobacco plants.

3. Subcellular localization

- Please indicate the literature reference or catalog number/accession number of the different organelle markers used. If plasmids were constructed ad-hoc for this work, please describe their construction. The authors might consider including a table of plasmids used.

- “For the localization of AtALA1, roots of EGFP::AtALA1 transgenic Arabidopsis were stained by FM4-64 (4 μ M, F34653, Thermo Scientific) in EGFP-AtALA1 transgenic Arabidopsis.” Please rephrase

- Please indicate what type of objective was used for the microscopy experiments and add the excitation and recording wavelengths for each fluorophore, as well as the pinhole aperture for each experiment. Were all images generated under identical pinhole apertures and with the same magnification?

4. Microscopy observation

Please specify in the title that this is referring to a set of specific experiments

Reviewer #3 (Remarks to the Author):

The authors identified Arabidopsis P4-ATPase genes, AtALA1 and AtALA7, responsible for the cell detoxification of the mycotoxins

produced by *Fusarium graminearum* and *Verticillium dahlia*. They showed that by AtALA1-/AtALA7-mediated vesicle transport,

the toxins were compartmentalized in vacuoles for degradation. They further proved that overexpression of AtALA1 and AtALA7

significantly increased the resistance of transgenic Arabidopsis and maize plants to *F. graminearum* and *V. dahlia*.

Novel data are well presented and analyzed. Except there are many language and grammatic issues which are pointed out by

the reviewer in the PDF file attached. The authors are encouraged to correct and edit these problems.

Reviewer #1 (Remarks to the Author):

Wang et al. in the manuscript entitled "Arabidopsis P4 ATPase-mediated cell detoxification confers the resistance to *Fusarium graminearum* and *Verticillium dahliae*" argue that ALA1 and ALA7 detoxify fungal toxins, DON and CIA, by trafficking them into vacuoles. All data are likely to support the authors' claim, but more precise experimental results should be presented to be published in the journal. In addition, the authors are likely to pick or select only a part of data to support their argument.

1. To show cellular localization of ALA1 and ALA7, the authors generated transgenic Arabidopsis plants expressing GFP or RFP-fused proteins by 35S promoter. Using these plants or sometimes by heterologous expression in tobacco with some plant organellar markers, the authors argue that ALA1/7 are transported from the PM via endosomes to vacuole. However, as the authors may know, by overexpression proteins can be localized to additional cellular compartments to their original sites. Therefore, the authors should generate and use plants expressing fusion proteins by native promoters to precisely observe their localization in plant cells.

Thanks! It's a good idea to use native promoter to control the GFP or RFP-fused proteins to reduce the mis-localization from overexpression. As suggested, we generated transgenic Arabidopsis in which the fusion proteins were under control of native promoters (*proAtALA1::EGFP-AtALA1* and *proAtALA7::AtALA7-eGFP*). The new data were presented in Fig 1.

2. The authors found that DON or CIA are not accumulated in FM4-64-stained vacuoles in *ala1* or *ala7* mutant in Fig. 1. But, DON and CIA induces leaf whitening in plants, which is accelerated in the mutant plants, as shown in the same Fig. 1. This may mean that cell death progress is more rapidly induced in those mutants by DON and CIA. Therefore, the authors have to show the localization of DON and CIA at earlier time points as Fig. 3b, or by movie in mutant plants together with WT.

According Tian and colleagues' observation, at 24 h after treating cotton plants with VD-toxins, there was no DNA fragmented ladders, a common characteristics of PCD, emerged on agarose gel; the DNA ladders were observed at 36 h [see Fig7, page 8, "Expression of Baculovirus Anti-Apoptotic Genes p35 and *op-iap* in Cotton (*Gossypium hirsutum* L.) Enhances Tolerance to *Verticillium Wilt*", PLoS ONE, 2010, 5 (12), e14218]. For *Fusarium* toxins, based on the observation of Masuda D., and colleagues, the cytological damage induced by DON was observed at 7 days after transferred to MS supplied with DON ($10 \mu\text{mol}\cdot\text{L}^{-1}$) [Phytotoxic effects of trichothecenes on the growth and morphology of *Arabidopsis thaliana*. J Exp Bot. 2007, 58 (7): 1617-1626].

To avoid the possibility that the failure of the accumulation of CIA and DON in vacuoles is resulted from the PCD, we began the observation of the distribution of the toxins at earlier time (for DON, at 2h; for CIA at 1h). Based on our experiments, the earliest time window for observation of the appearance of CIA in pre-vacuoles is at ~2 hours after feeding the plant with CIA^{FITC}; for DON, the time is ~6h. We believe that there's not enough time for CIA (~2h) and DON (~6h) to damage the transport and vacuole compartmentation of the toxins. The distribution of DON and CIA at earlier time points was added in Fig. 1d and i.

3. To test disease resistance to fungi, the authors generated transgenic Arabidopsis plants overexpressing ALA1 or ALA7 only. I wonder whether transgenic plant overexpressing fluorescently tagged proteins also show similar enhanced resistance to tested fungi? Are there any specific reasons to use ALA1 or ALA7 only to generate transgenic plants for resistance test?

Yes, the transgenic Arabidopsis overexpressing fluorescently tagged proteins also show similar enhanced resistance to tested fungi with native protein (see image below). To avoid possible negative impact from GFP fusion on the function of the proteins when expressed in other plants (e.g., tobacco and maize), we used the native AtALA1, instead of EGFP-AtALA1 in transgenic maize.

a Verticillium -wilt symptoms of wild type, *ala7* mutant, *AtALA7* overexpressing *Arabidopsis* lines (35S::AtALA7 lines OE-9, OE-12; 35S::AtALA7-eGFP). Plants were inoculated with *V. dahliae* strain L2-1 (3 mL, 2×10^8 mL⁻¹ of conidia) for ~3 weeks. Scale bar, 2 cm. b Disease index. Data were shown in dots of three replicates (at least 15 plants each). Different letters represent significant differences at P < 0.05 by one-way ANOVA with Tukey multiple comparisons test.

4. The authors show leaf whitening (resistance to DON and CIA), root length (ALA7 overexpression) or plant weight (ALA1 overexpression) depending on an experiment. Are there any reasons for this? The authors should show all three kinds of results regardless of different experimental approaches.

We found that 1 μg of DON (mL^{-1}) can significantly inhibit the growth of wild-type roots. However, to bleach the leaves, the concentration should be 50 $\mu\text{g}\cdot\text{mL}^{-1}$. Therefore, we used different concentration of DON in different experiments. Same to the CIA.

5. In many figures, arrowheads are not clearly indicated.

Thanks, we replaced the arrowheads with better one.

6. Pictures in Fig. 2b and c were obtained at the same time point after DON treatment. Why DON is distinctly localized? In addition, why localization patterns in Fig. 2e are not observed in Fig. 2d?

Fig. 2b shows the DON distribution in root hair in which the structure of EE can be clearly observed, while Fig. 2c shows the DON distribution in the cells of root elongation region, where the mature vacuoles can be easily observed. Root hairs are highly polarized tubular extensions, in which polar organization of the cytoplasm and its organelles is highly required (see image below) [Ovecka et al., Endocytosis and vesicle trafficking during tip growth of root hairs. Protoplasma. 2005 Oct; 226(1-2):39-54]. The tonoplast localization of EGFP-AtALA1 could be also observed in root hair cells, and the image was added in Fig. 2e of the revised manuscript.

Fig. 2e shows the PM and tonoplast localization of AtALA1 in root hair cells, while Fig 2d shows the PM localization of AtALA1 in the cells of root elongation zone. Since Fig 2e is capable of showing the localization of AtALA1 both in PM and in tonoplast, Fig 2 d seems a bit redundant. Therefore, we deleted the image in the revised manuscript.

EGFP-AtALA1 was localized to PM and membranes of vesicle, EE, LE, and tonoplast, which was indicated by FM4-64 (4 μM) staining. EGFP-AtALA1, EGFP-AtALA1 transgenic Arabidopsis. The gene was under control of the native promoter of *AtALA1* (*proAtALA1*). Scale bar, 10 μm .

Reviewer #2 (Remarks to the Author):

The manuscript presents the putative role of two P4 ATPases (ALA1 and ALA7) in detoxification of fungal toxins. P4 ATPases are lipid translocators involved in vesicle formation, lipid signaling and generation of an asymmetry lipid distribution that can be used to recruit proteins to the surface of the membrane. Based on KO mutants, protein overexpression and the use of secretory pathway inhibitors, the authors demonstrate that overexpression of arabidopsis ALA proteins can be used to increase the resistance of several plant species to fungal pathogens. While their findings are interesting, the mechanism behind the observed phenotype are not clear and several points require attention.

General comments

1. The English language relatively good, but there are many paragraphs where the grammatical constructions get too complicated or the use of the language is not correct. E.g. "This strategy enables hosts to be prevented from the toxicity of the mycotoxins" or "Recently, it was turned out that..."- The authors should consider professional language editing.

According to the suggestion, we carefully edited the language.

2. While I am not a statistics expert, I am not sure the statistical analysis is correct for most of the figures. In most figures, the authors present a Student's t-test, which is meant to compare two means. However, most experiments include three or more means, as there is always a control and at least two mutant/complemented lines. I believe the correct way to do this analysis is using a test of the type of ANOVA, which compares several different means, even if the authors are only interested in showing the difference between each mutant and the wild type. This type of analysis has been used for Sup. Fig. 2, for instance.

We appreciate this suggestion. Accordingly, we reanalyzed our data using one-way ANOVA with Tukey multiple comparisons test.

3. In general, it is difficult to assess the quality of the bioimaging work, such as the subcellular localizations and the secretory pathway inhibitor experiments, as no mention is made of number of repetitions and no quantifications are provided in most cases. The authors should clearly state how many repetitions of each experiment were made and provide quantifications and statistical analyses.

We added the descriptions of how many repetitions of each experiment, and provided quantifications and statistical analyses in the legends.

Introduction

1. "The P4 subfamily of P-type ATPases (P4 ATPases) functions as phospholipid flippases that translocate specific phospholipid substrates from the exoplasmic/luminal leaflet to the cytoplasmic leaflet of biological membranes,..." All plant P4 ATPases characterized to date and most P4 ATPases from other organisms are phospholipid transporters, but several P4 ATPases from humans and yeast have now been shown to transport lipids not containing a phosphate group, so please correct his sentence to account for that fact.

Thanks a lot for the suggestion. We have adapted the description.

2. When describing the general characteristics and physiological function of P4 ATPases in different organisms the authors cite a number of reviews published from 1996 to 2016 (refs. 52-60), several of which are too old to reflect our knowledge of the proteins. In the past 4-5 years the field has made huge advances and several updated reviews have been published, including some specifically devoted to plant P4 ATPases. While some of the references could stay, at least the oldest one should be updated. In addition, the authors should avoid using original citations for the general subjects or they should otherwise be fair and cite all the original papers on the subject. Some suggestions for the authors:

López-Marqués, R.L. et al. (2021) Dynamic membranes: the multiple roles of P4 and P5 ATPases. *Plant Physiol.* DOI: 10.1093/plphys/kiaa065

Shin, H.-W. and Takatsu, H. (2019) Substrates of P4 -ATPases: beyond aminophospholipids (phosphatidylserine and phosphatidylethanolamine). *FASEB J.* 33, 3087–3096

Best, J.T. et al. (2019) Phospholipid flippases in membrane remodeling and transport carrier biogenesis. *Curr. Opin. Cell Biol.* 59, 8–15

Nintemann, S.J. et al. (2019) Catch you on the flip side: A critical review of flippase mutant phenotypes. *Trends Plant Sci.* 24, 468–478

Andersen, J.P. et al. (2016) P4-ATPases as phospholipid flippases—Structure, function, and enigmas. *Front. Physiol.* 7, 275

We appreciate this constructive criticism. Following the suggestion, we added some new references and removed some old ones.

3. When citing the role of plant P4 ATPases in plant development, one reference is missing:

Davis, J.A. et al. (2020) The lipid flippases ALA4 and ALA5 play critical roles in cell expansion and plant growth. *Plant Physiol.* 182, 2111–2125

We added the references in the modified manuscript. Thanks!

4. In the list of physiological roles of plant P4 ATPases "...temperature

tolerance⁶¹⁻⁶³, disease resistance⁶⁴⁻⁶⁶, auxin polar distribution⁶⁷, plant growth and development,...”, I suggest that the authors also include plant fertility, as several P4 ATPases are important for this function

McDowell, S.C. et al. (2013) Loss of the Arabidopsis thaliana P4-ATPase ALA3 reduces adaptability to temperature stresses and impairs vegetative, pollen, and ovule development. PLoS One 8, e62577

McDowell, S.C. et al. (2015) Loss of the Arabidopsis thaliana P4-ATPases ALA6 and ALA7 impairs pollen fitness and alters the pollen tube plasma membrane. Front. Plant Sci. 6, 197

Zhou, Y. et al. (2020) The tip-localized phosphatidylserine established by Arabidopsis ALA3 is crucial for Rab GTPase-mediated vesicle trafficking and pollen tube growth. Plant Cell 32, 3170–3187

We had adopted these references.

Results

1. The authors show that *ala1* and *ala7* mutants have defective accumulation of fungal toxins in their vacuoles and that this phenotype can be complemented by the expression of *atALA1* and *AtALA7*, respectively. Therefore the authors conclude: “These data reveal that, in Arabidopsis, *AtALA1* is responsible for the DON vacuole accumulation and *AtALA7* is for the CIA vacuole accumulation”.

In order to make this conclusion, the authors are lacking at least one additional KO line for each ALA that demonstrates that the phenotype is directly related to the ALA protein. While complementation shows that the ALA proteins can complement the phenotypes observed, the authors are overexpressing the proteins all over the plants using a 35S promoter. Thus, even if the phenotypes are not directly related to the deletion of ALA1 or ALA7 but to another gene deleted in the background, the ALA proteins might have an activity that can, when overexpressed, rescue the phenotype.

We agree with the reviewer that we need more KO lines for each ALA to demonstrate the reliability of the phenotypes. Accordingly, we selected two mutants of each ALA for analysis: for ALA1, they are *ala1-8* (salk_002106) and *ala1* (salk_056947); for ALA7, they are *ala7* (salk_125598) and *ala7-24* (salk_063917). The results were added in Fig 1.

2. “DON5-FAM signal appeared in RFP-RabA1d-indicated EE-like structures (Fig. 2b); the signal was finally converged in vacuoles which were surrounded by the γ -Tip (Fig. 2c).”

The authors do not make a time-course of the accumulation of fluorescent DON, so they should not make any statement that refers to this. In this case, “finally” implies that they are showing the substance travelling through EE to the vacuole with time, which is not the case.

Thanks. It was deleted.

3. "In Arabidopsis root cells, EGFP-AtALA1 localized at the PM and tonoplast, which was stained by FM4-64 (Fig. 2d,e)"

The localization of the protein with the tonoplast is not at all clear. The authors present a picture in which FM4-64, which clearly colocalizes with the tonoplast does not really overlap with the signal for ALA1. ALA1 seems to be present in vesicular structures surrounding the vacuole (perhaps prevacuolar compartments) and a diffuse membrane surrounding the vacuole (ER?), but there is no clear tonoplast colocalization. This is supported by the low Rp value.

We observed the localization of AtALA1 in *proAtALA1::EGFP-AtALA1* transgenic Arabidopsis, and replaced the images with new one, in which the tonoplast colocalization of AtALA1 can be clearly observed (also see response to review1, question 6).

4. "In tobacco epidermal cells, EGFP-AtALA1 was colocalized with RFP-RabF2a (Fig. 2g)."

It has been shown before that ALA1 does not leave the ER in tobacco epidermal cells in the absence of a co-expressed beta subunit (López-Marqués, R.L. et al. (2012) A putative plant aminophospholipid flippase, the Arabidopsis P4 ATPase ALA1, localizes to the plasma membrane following association with a β -subunit. PLoS One 7, e33042). When co-expressed with a beta subunit, ALA1 travels to the plasma membrane and no association with endomembranes was observed in these experiments. How can the authors explain this discrepancy?

To address this problem, we co-expressed *proAtALA1::EGFP-AtALA1* with a β -subunit, ALIS1, in tobacco. As described by López-Marqués, R.L. et al. [A putative plant aminophospholipid flippase, the Arabidopsis P4 ATPase ALA1, localizes to the plasma membrane following association with a β -subunit. PLoS ONE, 2012, 7, e33042.], AtALA1 needs β -subunit to exit from ER. The data were added in Supplementary Fig.7.

To confirm the PM localization of AtALA1, we constructed *proAtALA1::EGFP-AtALA1* transgenic Arabidopsis. Similar to *35S::EGFP-AtALA1* Arabidopsis, the EGFP-AtALA1 signal appeared in the PM. We think the discrepancy may be from the different transformation methods: for the stable ectopic expression of EGFP-AtALA1 (i.e., *ProAtALA1::EGFP-AtALA1*) in Arabidopsis, the endogenous ALISs can help ALA1 traveling from ER to the plasma membrane; for the transient expression in tobacco, the trafficking needs the help of the co-expression of its binding partner, ALIS.

5. "In the BFA and DON5-FAM treated cells, massive aggregates of DON5-FAM and BFA induced FM4-64 aggregates were formed, but the DON5-FAM signal

was not colocalized to the BFA bodies (Fig. 2h)."

Can the authors rule out that what they call "massive aggregates" are not simply an accumulation of DON in the vacuoles? Trafficking from the Golgi is affected by BFA, but the authors have not shown so far that the Golgi is involved in trafficking of DON to the vacuole. In fact, the signals obtained for BFA treatment for CIA-treated plants are clearly different than those obtained for DON, which suggests the two substances do not follow the same route to the vacuole.

Our data show that the entry speed of CIA into cytosol is faster than that of DON (Supplementary Fig. 3e,f). The BFA treated time for DON^{5-FAM} in Fig. 2h was 12 h, while that for CIA^{FITC} in Fig. 3d was 3 h. To address this problem that "In fact, the signals obtained for BFA treatment for CIA-treated plants are clearly different than those obtained for DON", we prolonged the observing time for CIA from three hours previously to six hours. The distribution of CIA^{FITC} signal under BFA treatment was similar to that of DON signal (see image below). We replaced the imaging in Fig 3 d with new one.

Yes, we do not show that the Golgi is involved in trafficking of DON to the vacuole. Both DON and CIA signal are independent of BFA body, confirming that the traffic of DON and CIA to vacuoles is independent of Golgi.

6. The subcellular localization of ALA7 with GFP (Figure 3e) is not convincing. The ALA7 signal looks like the ER membrane, which is attached to the plasma membrane through contact points even after plasmolysis. To date, no ALA protein has been shown to be able to leave the ER in the absence of a beta subunit. To make these experiments credible (both for ALA1 and ALA7), the tobacco localization needs to be repeated in the presence of a beta subunit.

We appreciate this suggestion. Accordingly, we conducted co-expression of

proAtALA7::AtALA7-eGFP with a β -subunit, ALIS1. As you pointed out, AtALA7 indeed needs β -subunit to exit from ER (Supplementary Fig.7).

7. Do *ala1* or *ala7* plants show any cellular phenotypes in vesicular trafficking that can support their role? All the experiments are based on protein overexpression and there is no direct evidence of the involvement of any of the two ALA proteins in vesicle formation under native conditions.

Yes, we agree with that although we observe the bubble-like formation containing signals of the two proteins and two toxins emerged on the PM (Fig 1d; Fig 1i and 3b), details about the ALA proteins in vesicle formation are required. Nevertheless, the dual labeling of ALA proteins and their transport cargos described here allows us to observe the real time process of cargo trafficking as well as dynamic alteration of vesicles, EEs, LEs, and vacuoles *in vivo*, thus providing a platform to study the subcellular processes of P4-ATPase mediated vesicle transport and the important aspects of the compartmental detoxification in living cells.

Discussion

1. "With amphophilic structure, in the contrary, the ability of DON to enter the cells through diffusing across ..."

Amphophilic means "having an affinity for both acid and basic dyes." Do the authors mean "amphiphilic" (having both hydrophilic and hydrophobic parts)? The term is used repeatedly in the text.

Yes, we mean DON is an amphiphilic compound. Thank you for point out this mistaken.

2. "Possibly, AtALA1 is responsible for transport of some amphophilic molecules like DON; while AtALA7 is responsible for that of lipophilic compounds as CIA." The authors have not provided evidence that their results can be generalized like this to other molecules, and should tone down this statement.

In our previous study, we identified a P4-ATPase, BbCrpa, from entomopathogenic fungus, *Beauveria bassiana*. We show that BbCrpa is capable of transporting cyclosporine A (CsA, a lipophilic cyclic polypeptide), tacrolimus (FK506), as well as CIA. All these three compounds are lipophilic. This finding promotes us to identify the functional homologues of BbCrpa in Arabidopsis. In this study, we found ALA7 is responsible for transport of CIA, while ALA1 is for amphiphilic DON. The results from BbCrpa and ALA7/1 encourage us to propose this hypothesis. Of course, this hypothesis needs further study, which is our next research focusing on. According to the comment, in the revised manuscript, the hypothetical statement was weakened.

3. The authors seem to have forgotten to write a reference to Sup. Fig. 7 in the text. In this figure, they present a model of how ALA proteins might be involved in the observed detoxification effects. As mentioned in my comments above, the BFA treatment of plants suggests slightly different transport pathways to the vacuole for both substances and a putative model should reflect these differences.

Thank you for pointing out this omission. We added the reference to the modified text. By the way, we move the model from Supplementary into the text (Fig. 7). We observed the BFA treatment again and did not find significant difference between DON and CIA-treated cells.

4. The role of ALAs in detoxification of fungal toxins has not yet been described, but a role on detoxification of heavy metals was found for ALA4 (Sanz-Fernández, M. et al. (2017) Screening Arabidopsis mutants in genes useful for phytoremediation. J. Hazard. Mater. 335, 143–151). The authors could use this as an argument for the general use of flippases in detoxification.

Thank you for your suggestion. We emphasized the use of flippases in mycotoxin detoxification in the section of Introduction.

Figures

Sup. Figure 2: For the treatment with CIA there is not quantification of the amount of cells containing fluorescent CIA. Please include this information. The co-localization of signals along an axis is not giving any more information that what can be easily seen in the pictures and does not provide any information on whether the pictures are representative of a population.

Thanks! Accordingly, we added the quantification of the amount of cells containing fluorescent CIA in the figure.

Sup. Figure 3: Panel f, 6 hours. The image is not really showing a vacuolar localization of the fluorescent DON and it looks very much like bleed through signal.

We had replaced the image with better one.

Figure 3, panel d, Wm- The image for the fluorescent CIA is very poor. The authors should consider showing an image that does not look like background signal and clearly shows the fluorescence is inside the cells, if it is present there.

Thanks! We repeated the observation and replaced the image with better one.

Methods

1. In "Plant Materials and Growth Conditions" please indicate the growth conditions for tobacco plants as well.

It had been added.

2. In the generation of transgenic plants, please include the procedure for tobacco plants.

It had been done.

3. Subcellular localization

- Please indicate the literature reference or catalog number/accession number of the different organelle markers used. If plasmids were constructed ad-hoc for this work, please describe their construction. The authors might consider including a table of plasmids used.

A table of plasmids had been supplied in the revised manuscript (Supplementary Table 3).

- "For the localization of AtALA1, roots of EGFP::AtALA1 transgenic Arabidopsis were stained by FM4-64 (4 μ M, F34653, Thermo Scientific) in EGFP-AtALA1 transgenic Arabidopsis.". Please rephrase

It had done.

- Please indicate what type of objective was used for the microscopy experiments and add the excitation and recording wavelengths for each fluorophore, as well as the pinhole aperture for each experiment. Were all images generated under identical pinhole apertures and with the same magnification?

Thanks! It had been done.

4. Microscopy observation

Please specify in the title that this is referring to a set of specific experiments

Thanks! It had been done.

References cited:

López-Marqués, R.L., Poulsen, L.R., and Palmgren, M.G. (2012). A putative plant aminophospholipid

flippase, the Arabidopsis P4 ATPase ALA1, localizes to the plasma membrane following association with a β -subunit. PLoS ONE 7, e33042.

Masuda, D., Ishida, M., Yamaguchi, K., Yamaguchi, I., Kimura, M., and Nishiuchi, T. (2007). Phytotoxic effects of trichothecenes on the growth and morphology of Arabidopsis thaliana. J Exp Bot 58, 1617-1626.

Tian, J., Zhang, X., Liang, B., Li, S., Wu, Z., Wang, Q., Leng, C., Dong, J., and Wang, T. (2010). Expression of baculovirus anti-apoptotic genes p35 and op-iap in cotton (Gossypium hirsutum L.) enhances tolerance to verticillium wilt. PLoS ONE 5, e14218.

Reviewer #3 (Remarks to the Author):

The authors identified Arabidopsis P4-ATPase genes, AtALA1 and AtALA7, responsible for the cell detoxification of the mycotoxins produced by Fusarium graminearum and Verticillium dahlia. They showed that by AtALA1-/AtALA7-mediated vesicle transport, the toxins were compartmentalized in vacuoles for degradation. They further proved that overexpression of AtALA1 and AtALA7 significantly increased the resistance of transgenic Arabidopsis and maize plants to F. graminearum and V. dahlia. Novel data are well presented and analyzed. Except there are many language and grammatic issues which are pointed out by the reviewer in the PDF file attached. The authors are encouraged to correct and edit these problems.

Thank you very much for your editing. According the suggestion, we carefully correct and edit these problems.

REVIEWER COMMENTS

Reviewer #1 (Remarks to the Author):

The authors well-addressed all my concerns raised on the previous version of manuscript. I therefore think that this revised manuscript can be published in the journal. I also think that the trafficking-based detoxification of fungal toxins would attract broad interest from plant scientists, especially in the field of plant immunity.

Reviewer #2 (Remarks to the Author):

The manuscript presents the putative role of two P4 ATPases (ALA1 and ALA7) in detoxification of fungal toxins. P4 ATPases are lipid translocators involved in vesicle formation, lipid signaling and generation of an asymmetry lipid distribution that can be used to recruit proteins to the surface of the membrane. Based on KO mutants, protein overexpression and the use of secretory pathway inhibitors, the authors demonstrate that overexpression of arabidopsis ALA proteins can be used to increase the resistance of several plant species to fungal pathogens. While the manuscript has clearly improved from the last version, several points still require attention.

1. The English language is still relatively poor in the discussion section and at certain points in the introduction. Modified figure legends, such as that of Fig. 1 also require attention.
2. The authors have included additional KO lines for ALA1 and ALA7 in Fig. 1. However, they have failed to use these lines in the analyses in Fig. 4 and Fig-5. These KO lines should be consistently used to support the results
3. In Fig. 1d, the authors claim that the signal is observed in prevacuolar compartments. How is the identity of the observed organelles assigned? The morphology of vacuoles in plants is dependent on development, cells type, environmental conditions,... If the authors want to claim these are prevacuolar compartments and not a fragmented vacuole, they need to present evidence of the nature of this compartment.
4. There is still a discrepancy between figure 1d and Sup. Fig. 3f. While fig. 1d presents a clear vacuolar localization of DON-5FAM after 6h and 10 h for the wild type, Sup. Fig. 3f shows a completely different localization at 6 h and 12h. The authors need to specify clearly the type of cells they are visualizing. In addition, the text does not make any reference to the different vacuolar-compartmentalization behaviors in different types of cells, and this should be clearly stated.
5. In Figure 2, Rp and Rs coeficients are presented as a proof of colocalization. The authors do nor present any statistical analysis or confirmation that these calculations were based on more than one measurement. Please indicate the number of repetitions and include deviations.

REVIEWER COMMENTS

Reviewer #1 (Remarks to the Author):

The authors well-addressed all my concerns raised on the previous version of manuscript. I therefore think that this revised manuscript can be published in the journal. I also think that the trafficking-based detoxification of fungal toxins would attract broad interest from plant scientists, especially in the field of plant immunity.

Thank you for these positive comments.

Reviewer #2 (Remarks to the Author):

The manuscript presents the putative role of two P4 ATPases (ALA1 and ALA7) in detoxification of fungal toxins. P4 ATPases are lipid translocators involved in vesicle formation, lipid signaling and generation of an asymmetry lipid distribution that can be used to recruit proteins to the surface of the membrane. Based on KO mutants, protein overexpression and the use of secretory pathway inhibitors, the authors demonstrate that overexpression of arabidopsis ALA proteins can be used to increase the resistance of several plant species to fungal pathogens. While the manuscript has clearly improved from the last version, several points still require attention.

1. The English language is still relatively poor in the discussion section and at certain points in the introduction. Modified figure legends, such as that of Fig. 1 also require attention.

Thanks. We checked the English language and revised the manuscript.

2. The authors have included additional KO lines for ALA1 and ALA7 in Fig. 1. However, they have failed to use these lines in the analyses in Fig. 4 and Fig-5. These KO lines should be consistently used to support the results.

We appreciate this advice. As suggested, we added new KO lines of ALA1 and ALA7 in Fig 1, and re-performed all experiments in Fig.4 and Fig.5.

3. In Fig. 1d, the authors claim that the signal is observed in prevacuolar compartments. How is the identity of the observed organelles assigned? The morphology of vacuoles in plants is dependent on development, cells type, environmental conditions,... If the authors want to claim these are prevacuolar compartments and not a fragmented vacuole, they need to present evidence of the nature of this compartment.

Thanks for this critical comment. According to the report of Cui et al [Cui, Y., Cao, W., He, Y., Zhao, Q., Wakazaki, M., Zhuang, X., ... & Jiang, L. (2019). A whole-cell electron tomography model of vacuole biogenesis in Arabidopsis root cells. Nature plants, 5(1), 95-105], the small

vacuoles (SVs; 400–1,000 nm in diameter) are the major source for the larger vacuole formation in these early developmental stage cells. In Fig. 1d, the organelles diameter observed at 6 h were 500-8000 nm, it may contain SVs and larger vacuole. Accordingly we replaced prevacuolar compartments with small vacuoles.

4. There is still a discrepancy between figure 1d and Sup. Fig. 3f. While fig. 1d presents a clear vacuolar localization of DON-5FAM after 6h and 10 h for the wild type, Sup. Fig. 3f shows a completely different localization at 6 h and 12h. The authors need to specify clearly the type of cells they are visualizing. In addition, the text does not make any reference to the different vacuolar-compartmentalization behaviors in different types of cells, and this should be clearly stated.

We think the discrepancy may be from the different DON^{5-FAM} concentration. In Fig.1 d, we used DON^{5-FAM} at concentration of $9 \mu\text{g}\cdot\text{mL}^{-1}$ to compare the distribution DON among mutants and wild type, while in Sup. Fig. 3f, we used equal concentration of DON and CIA to compare their entry speed, which is much lower ($4 \mu\text{g}\cdot\text{mL}^{-1}$) than that of the DON in Fig. 1 ($9 \mu\text{g}\cdot\text{mL}^{-1}$). At higher concentration, the DON signal was much stronger than that under lower one. That is why a clear vacuolar localization of DON^{5-FAM} can be observed after 6h and 10 h in Fig. 1d, but not in Sup. Fig. 3e-f.

5. In Figure 2, Rp and Rs coefficients are presented as a proof of colocalization. The authors do not present any statistical analysis or confirmation that these calculations were based on more than one measurement. Please indicate the number of repetitions and include deviations.

Thanks for the valuable suggestions. In the revised manuscript, we added statistical data with mean \pm SD in Fig. 2 and Fig-3. For the number of repetitions, the co-localization assays were independently repeated at least three times (see the legends of Fig. 2 and Fig.3).

REVIEWERS' COMMENTS

Reviewer #2 (Remarks to the Author):

The authors have addressed my concerns appropriately.